# TOWARDS STABLE AND EFFICIENT TRAINING OF VERIFIABLY ROBUST NEURAL NETWORKS

**Huan Zhang**[1]* **Hongge Chen**[2] **Chaowei Xiao**[3] **Sven Gowal**[4] **Robert Stanforth**[4]
**Bo Li**[5] **Duane Boning**[2] **Cho-Jui Hsieh**[1]
[1] UCLA  [2] MIT  [3] University of Michigan  [4] DeepMind  [5] UIUC
huan@huan-zhang.com, chenhg@mit.edu, xiaocw@umich.edu
sgowal@google.com, stanforth@google.com
lbo@illinois.edu, boning@mtl.mit.edu, chohsieh@cs.ucla.edu

## ABSTRACT

Training neural networks with verifiable robustness guarantees is challenging. Several existing approaches utilize linear relaxation based neural network output bounds under perturbation, but they can slow down training by a factor of hundreds depending on the underlying network architectures. Meanwhile, interval bound propagation (IBP) based training is efficient and significantly outperforms linear relaxation based methods on many tasks, yet it may suffer from stability issues since the bounds are much looser especially at the beginning of training. In this paper, we propose a new certified adversarial training method, **CROWN-IBP**, by combining the fast IBP bounds in a forward bounding pass and a tight linear relaxation based bound, CROWN, in a backward bounding pass. CROWN-IBP is computationally efficient and consistently outperforms IBP baselines on training verifiably robust neural networks. We conduct large scale experiments on MNIST and CIFAR datasets, and outperform all previous linear relaxation and bound propagation based certified defenses in $\ell_\infty$ robustness. Notably, we achieve 7.02% verified test error on MNIST at $\epsilon = 0.3$, and 66.94% on CIFAR-10 with $\epsilon = 8/255$.

## 1 INTRODUCTION

The success of deep neural networks (DNNs) has motivated their deployment in some safety-critical environments, such as autonomous driving and facial recognition systems. Applications in these areas make understanding the robustness and security of deep neural networks urgently needed, especially their resilience under malicious, finely crafted inputs. Unfortunately, the performance of DNNs are often so brittle that even imperceptibly modified inputs, also known as adversarial examples, are able to completely break the model (Goodfellow et al., 2015; Szegedy et al., 2013). The robustness of DNNs under adversarial examples is well-studied from both attack (crafting powerful adversarial examples) and defence (making the model more robust) perspectives (Athalye et al., 2018; Carlini & Wagner, 2017a;b; Goodfellow et al., 2015; Madry et al., 2018; Papernot et al., 2016; Xiao et al., 2019b; 2018b;c; Eykholt et al., 2018; Chen et al., 2018; Xu et al., 2018; Zhang et al., 2019b). Recently, it has been shown that defending against adversarial examples is a very difficult task, especially under strong and adaptive attacks. Early defenses such as distillation (Papernot et al., 2016) have been broken by stronger attacks like C&W (Carlini & Wagner, 2017b). Many defense methods have been proposed recently (Guo et al., 2018; Song et al., 2017; Buckman et al., 2018; Ma et al., 2018; Samangouei et al., 2018; Xiao et al., 2018a; 2019a), but their robustness improvement cannot be *certified* – no provable guarantees can be given to verify their robustness. In fact, most of these uncertified defenses become vulnerable under stronger attacks (Athalye et al., 2018; He et al., 2017).

Several recent works in the literature seeking to give *provable* guarantees on the robustness performance, such as linear relaxations (Wong & Kolter, 2018; Mirman et al., 2018; Wang et al., 2018a; Dvijotham et al., 2018b; Weng et al., 2018; Zhang et al., 2018), interval bound propagation (Mirman et al., 2018; Gowal et al., 2018), ReLU stability regularization (Xiao et al., 2019c), and distributionally

---

*Work partially done during an internship at DeepMind.

robust optimization (Sinha et al., 2018) and semidefinite relaxations (Raghunathan et al., 2018a; Dvijotham et al.). Linear relaxations of neural networks, first proposed by Wong & Kolter (2018), is one of the most popular categories among these certified defences. They use the dual of linear programming or several similar approaches to provide a linear relaxation of the network (referred to as a "convex adversarial polytope") and the resulting bounds are tractable for robust optimization. However, these methods are both computationally and memory intensive, and can increase model training time by a factor of hundreds. On the other hand, interval bound propagation (IBP) is a simple and efficient method for training verifiable neural networks (Gowal et al., 2018), which achieved state-of-the-art verified error on many datasets. However, since the IBP bounds are very loose during the initial phase of training, the training procedure can be unstable and sensitive to hyperparameters.

In this paper, we first discuss the strengths and weakness of existing linear relaxation based and interval bound propagation based certified robust training methods. Then we propose a new certified robust training method, CROWN-IBP, which marries the efficiency of IBP and the tightness of a linear relaxation based verification bound, CROWN (Zhang et al., 2018). CROWN-IBP bound propagation involves a IBP based fast forward bounding pass, and a tight convex relaxation based backward bounding pass (CROWN) which scales linearly with the size of neural network output and is very efficient for problems with low output dimensions. Additional, CROWN-IBP provides flexibility for exploiting the strengths of both IBP and convex relaxation based verifiable training methods.

The *efficiency*, *tightness* and *flexibility* of CROWN-IBP allow it to outperform state-of-the-art methods for training verifiable neural networks with $\ell_\infty$ robustness under all $\epsilon$ settings on MNIST and CIFAR-10 datasets. In our experiment, on MNIST dataset we reach $7.02\%$ and $12.06\%$ IBP verified error under $\ell_\infty$ distortions $\epsilon = 0.3$ and $\epsilon = 0.4$, respectively, outperforming the state-of-the-art baseline results by IBP ($8.55\%$ and $15.01\%$). On CIFAR-10, at $\epsilon = \frac{2}{255}$, CROWN-IBP decreases the verified error from $55.88\%$ (IBP) to $46.03\%$ and matches convex relaxation based methods; at a larger $\epsilon$, CROWN-IBP outperforms all other methods with a noticeable margin.

## 2 RELATED WORK AND BACKGROUND

### 2.1 ROBUSTNESS VERIFICATION AND RELAXATIONS OF NEURAL NETWORKS

Neural network robustness verification algorithms seek for upper and lower bounds of an output neuron for all possible inputs within a set $S$, typically a norm bounded perturbation. Most importantly, the margins between the ground-truth class and any other classes determine model robustness. However, it has already been shown that finding the exact output range is a non-convex problem and NP-complete (Katz et al., 2017; Weng et al., 2018). Therefore, recent works resorted to giving relatively tight but computationally tractable bounds of the output range with necessary relaxations of the original problem. Many of these robustness verification approaches are based on linear relaxations of non-linear units in neural networks, including CROWN (Zhang et al., 2018), DeepPoly (Singh et al., 2019), Fast-Lin (Weng et al., 2018), DeepZ (Singh et al., 2018) and Neurify (Wang et al., 2018b). We refer the readers to (Salman et al., 2019b) for a comprehensive survey on this topic. After linear relaxation, they bound the output of a neural network $f_i(\cdot)$ by linear upper/lower hyper-planes:

$$\mathbf{A}_{i,:}\Delta\boldsymbol{x} + b_L \le f_i(\boldsymbol{x}_0 + \Delta\boldsymbol{x}) \le \mathbf{A}_{i,:}\Delta\boldsymbol{x} + b_U \tag{1}$$

where a row vector $\mathbf{A}_{i,:} = \mathbf{W}_{i,:}^{(L)}\mathbf{D}^{(L-1)}\mathbf{W}^{(L-1)}\cdots\mathbf{D}^{(1)}\mathbf{W}^{(1)}$ is the product of the network weight matrices $\mathbf{W}^{(l)}$ and diagonal matrices $\mathbf{D}^{(l)}$ reflecting the ReLU relaxations for output neuron $i$; $b_L$ and $b_U$ are two bias terms unrelated to $\Delta x$. Additionally, Dvijotham et al. (2018c;a); Qin et al. (2019) solve the Lagrangian dual of verification problem; Raghunathan et al. (2018a;b); Dvijotham et al. propose semidefinite relaxations which are tighter compared to linear relaxation based methods, but computationally expensive. Bounds on neural network local Lipschitz constant can also be used for verification (Zhang et al., 2019c; Hein & Andriushchenko, 2017). Besides these deterministic verification approaches, randomized smoothing can be used to certify the robustness of any model in a probabilistic manner (Cohen et al., 2019; Salman et al., 2019a; Lecuyer et al., 2018; Li et al., 2018).

### 2.2 ROBUST OPTIMIZATION AND VERIFIABLE ADVERSARIAL DEFENSE

To improve the robustness of neural networks against adversarial perturbations, a natural idea is to generate adversarial examples by attacking the network and then use them to augment the training set (Kurakin et al., 2017). More recently, Madry et al. (2018) showed that adversarial training can

be formulated as solving a minimax robust optimization problem as in (2). Given a model with parameter $\theta$, loss function $L$, and training data distribution $\mathcal{X}$, the training algorithm aims to minimize the robust loss, which is defined as the maximum loss within a neighborhood $\{x + \delta | \delta \in S\}$ of each data point $x$, leading to the following robust optimization problem:

$$\min_{\theta} \; E_{(x,y) \in \mathcal{X}} \left[ \max_{\delta \in S} L(x + \delta; y; \theta) \right]. \tag{2}$$

Madry et al. (2018) proposed to use projected gradient descent (PGD) to approximately solve the inner max and then use the loss on the perturbed example $x + \delta$ to update the model. Networks trained by this procedure achieve state-of-the-art test accuracy under strong attacks (Athalye et al., 2018; Wang et al., 2018a; Zheng et al., 2018). Despite being robust under strong attacks, models obtained by this PGD-based adversarial training do not have verified error guarantees. Due to the nonconvexity of neural networks, PGD attack can only compute the *lower bound* of robust loss (the inner maximization problem). Minimizing a lower bound of the inner max cannot guarantee (2) is minimized. In other words, even if PGD-attack cannot find a perturbation with large loss, that does not mean there exists no such perturbation. This becomes problematic in safety-critical applications since those models need *certified* safety.

Verifiable adversarial training methods, on the other hand, aim to obtain a network with good robustness that can be verified efficiently. This can be done by combining adversarial training and robustness verification—instead of using PGD to find a lower bound of inner max, certified adversarial training uses a verification method to find an *upper bound* of the inner max, and then update the parameters based on this upper bound of robust loss. Minimizing an upper bound of the inner max guarantees to minimize the robust loss. There are two certified robust training methods that are related to our work and we describe them in detail below.

**Linear Relaxation Based Verifiable Adversarial Training.** One of the most popular verifiable adversarial training method was proposed in (Wong & Kolter, 2018) using linear relaxations of neural networks to give an upper bound of the inner max. Other similar approaches include Mirman et al. (2018); Wang et al. (2018a); Dvijotham et al. (2018b). Since the bound propagation process of a convex adversarial polytope is too expensive, several methods were proposed to improve its efficiency, like Cauchy projection (Wong et al., 2018) and dynamic mixed training (Wang et al., 2018a). However, even with these speed-ups, the training process is still slow. Also, this method may significantly reduce a model's standard accuracy (accuracy on natural, unmodified test set). As we will demonstrate shortly, we find that this method tends to over-regularize the network during training, which is harmful for obtaining good accuracy.

**Interval Bound Propagation (IBP).** Interval Bound Propagation (IBP) uses a very simple rule to compute the pre-activation outer bounds for each layer of the neural network. Unlike linear relaxation based methods, IBP does not relax ReLU neurons and does not consider the correlations between neurons of different layers, yielding much looser bounds. Mirman et al. (2018) proposed a variety of abstract domains to give sound over-approximations for neural networks, including the "Box/Interval Domain" (referred to as IBP in Gowal et al. (2018)) and showed that it could scale to much larger networks than other works (Raghunathan et al., 2018a) could at the time. Gowal et al. (2018) demonstrated that IBP could outperform many state-of-the-art results by a large margin with more precise approximations for the last linear layer and better training schemes. However, IBP can be unstable to use and hard to tune in practice, since the bounds can be very loose especially during the initial phase of training, posing a challenge to the optimizer. To mitigate instability, Gowal et al. (2018) use a mixture of regular and minimax robust cross-entropy loss as the model's training loss.

## 3 METHODOLOGY

**Notation.** We define an $L$-layer feed-forward neural network recursively as:

$$f(\boldsymbol{x}) = z^{(L)} \qquad z^{(l)} = \mathbf{W}^{(l)} h^{(l-1)} + \boldsymbol{b}^{(l)} \qquad \mathbf{W}^{(l)} \in \mathbb{R}^{n_l \times n_{l-1}} \qquad \boldsymbol{b}^{(l)} \in \mathbb{R}^{n_l}$$

$$h^{(l)} = \sigma^{(l)}(z^{(l)}), \quad \forall l \in \{1, \dots, L-1\},$$

where $h^{(0)}(\boldsymbol{x}) = \boldsymbol{x}$, $n_0$ represents input dimension and $n_L$ is the number of classes, $\sigma$ is an element-wise activation function. We use $z$ to represent pre-activation neuron values and $h$ to represent

| Dataset | $\epsilon$ ($\ell_\infty$ norm) | CAP verified error | CROWN verified error | IBP verified error |
|---|---|---|---|---|
| MNIST | 0.1 | 8.90% | 7.05% | 5.83% |
| | 0.2 | 45.37% | 24.17% | 7.37% |
| | 0.3 | 97.77% | 65.26% | 10.68% |
| | 0.4 | 99.98% | 99.57% | 16.76% |
| Fashion-MNIST | 0.1 | 44.64% | 36.85% | 23.49% |
| CIFAR-10 | 2/255 | 62.94% | 60.83% | 58.75% |
| | 8/255 | 91.44% | 82.68% | 73.34% |

Table 1: IBP trained models have low IBP verified errors but when verified with a typically much tighter bound, including convex adversarial polytope (CAP) (Wong et al., 2018) and CROWN (Zhang et al., 2018), the verified errors increase significantly. CROWN is generally tighter than convex adversarial polytope however the gap between CROWN and IBP is still large, especially at large $\epsilon$. We used a 4-layer CNN network for all datasets to compute these bounds.[1]

post-activation neuron values. Consider an input example $\boldsymbol{x}_k$ with ground-truth label $y_k$, we define a set of $S(\boldsymbol{x}_k, \epsilon) = \{\boldsymbol{x} \| \|\boldsymbol{x} - \boldsymbol{x}_k\|_\infty \leq \epsilon\}$ and we desire a robust network to have the property $y_k = \mathrm{argmax}_j [f(\boldsymbol{x})]_j$ for all $\boldsymbol{x} \in S$. We define element-wise upper and lower bounds for $z^{(l)}$ and $h^{(l)}$ as $\underline{z}^{(l)} \leq z^{(l)} \leq \overline{z}^{(l)}$ and $\underline{h}^{(l)} \leq h^{(l)} \leq \overline{h}^{(l)}$.

**Verification Specifications.** Neural network verification literature typically defines a specification vector $\boldsymbol{c} \in \mathbb{R}^{n_L}$, that gives a linear combination for neural network output: $\boldsymbol{c}^\top f(\boldsymbol{x})$. In robustness verification, typically we set $\boldsymbol{c}_i = 1$ where $i$ is the ground truth class label, $\boldsymbol{c}_j = -1$ where $j$ is the attack target label and other elements in $c$ are 0. This represents the margin between class $i$ and class $j$. For an $n_L$ class classifier and a given label $y$, we define a specification matrix $C \in \mathbb{R}^{n_L \times n_L}$ as:

$$C_{i,j} = \begin{cases} 1, & \text{if } j = y, i \neq y \text{ (output of ground truth class)} \\ -1, & \text{if } i = j, i \neq y \text{ (output of other classes, negated)} \\ 0, & \text{otherwise (note that the } y\text{-th row contains all 0)} \end{cases} \quad (3)$$

Importantly, each element in vector $\boldsymbol{m} := Cf(\boldsymbol{x}) \in \mathbb{R}^{n_L}$ gives us margins between class $y$ and all other classes. We define the lower bound of $Cf(\boldsymbol{x})$ for all $\boldsymbol{x} \in S(\boldsymbol{x}_k, \epsilon)$ as $\underline{\boldsymbol{m}}(\boldsymbol{x}_k, \epsilon)$, which is a very important quantity: when all elements of $\underline{\boldsymbol{m}}(\boldsymbol{x}_k, \epsilon) > 0$, $\boldsymbol{x}_k$ is *verifiably robust* for any perturbation with $\ell_\infty$ norm less than $\epsilon$. $\underline{\boldsymbol{m}}(\boldsymbol{x}_k, \epsilon)$ can be obtained by a neural network verification algorithm, such as convex adversarial polytope, IBP, or CROWN. Additionally, Wong & Kolter (2018) showed that for cross-entropy (CE) loss:

$$\max_{\boldsymbol{x} \in S(\boldsymbol{x}_k, \epsilon)} L(f(\boldsymbol{x}); y; \theta) \leq L(-\underline{\boldsymbol{m}}(\boldsymbol{x}_k, \epsilon); y; \theta). \quad (4)$$

(4) gives us the opportunity to solve the robust optimization problem (2) via minimizing this tractable upper bound of inner-max. This guarantees that $\max_{\boldsymbol{x} \in S(\boldsymbol{x}_k, \epsilon)} L(f(\boldsymbol{x}), y)$ is also minimized.

### 3.1 ANALYSIS OF IBP AND LINEAR RELAXATION BASED VERIFIABLE TRAINING METHODS

**Interval Bound Propagation (IBP)** Interval Bound Propagation (IBP) uses a simple bound propagation rule. For the input layer we set $\boldsymbol{x}_L \leq \boldsymbol{x} \leq \boldsymbol{x}_U$ element-wise. For affine layers we have:

$$\overline{z}^{(l)} = \mathbf{W}^{(l)} \frac{\overline{h}^{(l-1)} + \underline{h}^{(l-1)}}{2} + |\mathbf{W}^{(l)}| \frac{\overline{h}^{(l-1)} - \underline{h}^{(l-1)}}{2} + b^{(l)} \quad (5)$$

$$\underline{z}^{(l)} = \mathbf{W}^{(l)} \frac{\overline{h}^{(l-1)} + \underline{h}^{(l-1)}}{2} - |\mathbf{W}^{(l)}| \frac{\overline{h}^{(l-1)} - \underline{h}^{(l-1)}}{2} + b^{(l)} \quad (6)$$

where $|\mathbf{W}^{(l)}|$ takes element-wise absolute value. Note that $\overline{h}^{(0)} = \boldsymbol{x}_U$ and $\underline{h}^{(0)} = \boldsymbol{x}_L$[2]. And for element-wise monotonic increasing activation functions $\sigma$,

$$\overline{h}^{(l)} = \sigma(\overline{z}^{(l)}) \qquad \underline{h}^{(l)} = \sigma(\underline{z}^{(l)}). \quad (7)$$

---

[1] We implemented CROWN with efficient CNN support on GPUs: https://github.com/huanzhang12/CROWN-IBP

[2] For inputs bounded with general norms, IBP can be applied as long as this norm can be converted to per-neuron intervals after the first affine layer. For example, for $\ell_p$ norms ($1 \leq p \leq \infty$) Hölder's inequality can be applied at the first affine layer to obtain $\overline{h}^{(1)}$ and $\underline{h}^{(1)}$, and IBP rule for later layers do not change.

We found that IBP can be viewed as training a simple augmented ReLU network which is friendly to optimizers (see Appendix A for more discussions). We also found that a network trained using IBP can obtain good verified errors when verified using IBP, but it can get much worse verified errors using linear relaxation based verification methods, including convex adversarial polytope (CAP) by Wong & Kolter (2018) (equivalently, Fast-Lin by Weng et al. (2018)) and CROWN (Zhang et al., 2018). Table 1 demonstrates that this gap can be very large on large $\epsilon$.

However, IBP is a very loose bound during the initial phase of training, which makes training unstable and hard to tune; purely using IBP frequently leads to divergence. Gowal et al. (2018) proposed to use a $\epsilon$ *schedule* where $\epsilon$ is gradually increased during training, and a mixture of robust cross-entropy loss with natural cross-entropy loss as the objective to stabilize training:

$$\min_{\theta} \; \mathop{E}_{(\boldsymbol{x},y)\in\mathcal{X}} \Big[ \kappa L(\boldsymbol{x};y;\theta) + (1-\kappa)L(-\underline{\boldsymbol{m}}_{\text{IBP}}(\boldsymbol{x},\epsilon);y;\theta) \Big], \tag{8}$$

**Issues with linear relaxation based training.** Since IBP hugely outperforms linear relaxation based methods in the recent work (Gowal et al., 2018) in many settings, we want to understand what is going wrong with linear relaxation based methods. We found that, empirically, the norm of the weights in the models produced by linear relaxation based methods such as (Wong & Kolter, 2018) and (Wong et al., 2018) does not change or even decreases during training.

In Figure 1 we train a small 4-layer MNIST model and we linearly increase $\epsilon$ from 0 to 0.3 in 60 epochs. We plot the $\ell_\infty$ induced norm of the 2nd CNN layer during the training process of CROWN-IBP and (Wong et al., 2018). The norm of weight matrix using (Wong et al., 2018) does not increase. When $\epsilon$ becomes larger (roughly at $\epsilon = 0.2$, epoch 40), the norm even starts to decrease slightly, indicating that the model is forced to learn smaller norm weights. Meanwhile, the verified error also starts to ramp up possibly due to the lack of capacity. We conjecture that linear relaxation based training over-regularizes the model, especially at a larger $\epsilon$. However, in CROWN-IBP, the norm of weight matrices keep increasing during the training process, and verifiable error does not significantly increase when $\epsilon$ reaches 0.3.

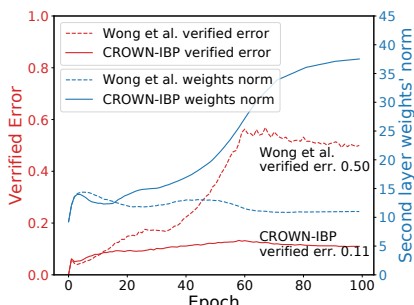

Figure 1: Verified error and 2nd CNN layer's $\ell_\infty$ induced norm for a model trained using (Wong et al., 2018) and CROWN-IBP. $\epsilon$ is increased from 0 to 0.3 in 60 epochs.

Another issue with current linear relaxation based training or verification methods is their high computational and memory cost, and poor scalability. For the small network in Figure 1, convex adversarial polytope (with 50 random Cauchy projections) is 8 times slower and takes 4 times more memory than CROWN-IBP (without using random projections). Convex adversarial polytope scales even worse for larger networks; see Appendix J for a comparison.

### 3.2 The Proposed Algorithm: CROWN-IBP

**Overview.** We have reviewed IBP and linear relaxation based methods above. As shown in Gowal et al. (2018), IBP performs well at large $\epsilon$ with much smaller verified error, and also efficiently scales to large networks; however, it can be sensitive to hyperparameters due to its very imprecise bound at the beginning phase of training. On the other hand, linear relaxation based methods can give tighter lower bounds at the cost of high computational expenses, but it over-regularizes the network at large $\epsilon$ and forbids us to achieve good standard and verified accuracy. We propose CROWN-IBP, a new certified defense where we optimize the following problem ($\theta$ represents the network parameters):

$$\min_{\theta} \; \mathop{E}_{(\boldsymbol{x},y)\in\mathcal{X}} \Big[ \kappa \underbrace{L(\boldsymbol{x};y;\theta)}_{\text{natural loss}} + (1-\kappa) \underbrace{L\big( -(\overbrace{(1-\beta)\underline{\boldsymbol{m}}_{\text{IBP}}(\boldsymbol{x},\epsilon)}^{\text{IBP bound}} + \overbrace{\beta\underline{\boldsymbol{m}}_{\text{CROWN-IBP}}(\boldsymbol{x},\epsilon)}^{\text{CROWN-IBP bound}}); \; y; \; \theta \big)}_{\text{robust loss}} \Big], \tag{9}$$

where our lower bound of margin $\underline{\boldsymbol{m}}(\boldsymbol{x},\epsilon)$ is a combination of two bounds with different natures: IBP, and a CROWN-style bound (which will be detailed below); $L$ is the cross-entropy loss. Note that the combination is inside the loss function and is thus still a valid lower bound; thus (4) still holds and we are within the minimax robust optimization theoretical framework. Similar to IBP and

TRADES (Zhang et al., 2019a), we use a mixture of natural and robust training loss with parameter $\kappa$, allowing us to explicitly trade-off between clean accuracy and verified accuracy.

In a high level, the computation of the lower bounds of CROWN-IBP ($\underline{m}_{\text{CROWN-IBP}}(\boldsymbol{x}, \epsilon)$) consists of IBP bound propagation in a forward bounding pass and CROWN-style bound propagation in a backward bounding pass. We discuss the details of CROWN-IBP algorithm below.

**Forward Bound Propagation in CROWN-IBP.** In CROWN-IBP, we first obtain $\overline{z}^{(l)}$ and $\underline{z}^{(l)}$ for all layers by applying (5), (6) and (7). Then we will obtain $\underline{m}_{\text{IBP}}(\boldsymbol{x}, \epsilon) = \underline{z}^{(L)}$ (assuming $C$ is merged into $\mathbf{W}^{(L)}$). The time complexity is comparable to two forward propagation passes of the network.

**Linear Relaxation of ReLU neurons** Given $\underline{z}^{(l)}$ and $\overline{z}^{(l)}$ computed in the previous step, we first check if some neurons are always active ($\underline{z}_k^{(l)} > 0$) or always inactive ($\overline{z}_k^{(l)} < 0$), since they are effectively linear and no relaxations are needed. For the remaining unstable neurons, Zhang et al. (2018); Wong & Kolter (2018) give a linear relaxation for ReLU activation function:

$$\alpha_k z_k^{(l)} \leq \sigma(z_k^{(l)}) \leq \frac{\overline{z}_k^{(l)}}{\overline{z}_k^{(l)} - \underline{z}_k^{(l)}} z_k^{(l)} - \frac{\overline{z}_k^{(l)} \underline{z}_k^{(l)}}{\overline{z}_k^{(l)} - \underline{z}_k^{(l)}}, \quad \text{for all } k \in [n_l] \text{ and } \underline{z}_k^{(l)} < 0 < \overline{z}_k^{(l)}, \quad (10)$$

where $0 \leq \alpha_k \leq 1$; Zhang et al. (2018) propose to adaptively select $\alpha_k = 1$ when $\overline{z}_k^{(l)} > |\underline{z}_k^{(l)}|$ and 0 otherwise, which minimizes the relaxation error. Following (10), for an input vector $z^{(l)}$, we effectively replace the ReLU layer with a linear layer, giving upper or lower bounds of the output:

$$\underline{\mathbf{D}}^{(l)} z^{(l)} \leq \sigma(z^{(l)}) \leq \overline{\mathbf{D}}^{(l)} z^{(l)} + \overline{c}_d^{(l)} \quad (11)$$

where $\underline{\mathbf{D}}^{(l)}$ and $\overline{\mathbf{D}}^{(l)}$ are two diagonal matrices representing the "weights" of the relaxed ReLU layer. Other general activation functions can be supported similarly. In the following we focus on conceptually presenting the algorithm, while more details of each term can be found in the Appendix.

**Backward Bound Propagation in CROWN-IBP.** Unlike IBP, CROWN-style bounds start bounding from the last layer, so we refer to it as *backward bound propagation* (not to be confused with the back-propagation algorithm to obtain gradients). Suppose we want to obtain the lower bound $[\underline{m}_{\text{CROWN-IBP}}(\boldsymbol{x}, \epsilon)]_i := \underline{z}_i^{(L)}$ (we assume the specification matrix $C$ has been merged into $\mathbf{W}^{(L)}$). The input to layer $\mathbf{W}^{(L)}$ is $\sigma(z^{(L-1)})$, which can be bounded linearly by Eq. (11). CROWN-style bounds choose the lower bound of $\sigma(z_k^{(L-1)})$ (LHS of (11)) when $\mathbf{W}_{i,k}^{(L)}$ is positive, and choose the upper bound otherwise. We then merge $\mathbf{W}^{(L)}$ and the linearized ReLU layer together and define:

$$\mathbf{A}_{i,:}^{(L-1)} = \mathbf{W}_{i,:}^{(L)} \mathbf{D}^{i,(L-1)}, \quad \text{where} \quad \mathbf{D}_{k,k}^{i,(L-1)} = \begin{cases} \underline{\mathbf{D}}_{k,k}^{(L-1)}, & \text{if } \mathbf{W}_{i,k}^{(L)} > 0 \\ \overline{\mathbf{D}}_{k,k}^{(L-1)}, & \text{if } \mathbf{W}_{i,k}^{(L)} \leq 0 \end{cases} \quad (12)$$

Now we have a lower bound $\underline{z}_i^{(L)} = \mathbf{A}_{i,:}^{(L-1)} z^{(L-1)} + \underline{b}_i^{(L-1)} \leq z_i^{(L)}$ where $\underline{b}_i^{(L-1)} = \sum_{k, \mathbf{W}_{i,k}^{(L)} < 0} \mathbf{W}_{i,k}^{(L)} \overline{c}_k^{(l)} + b^{(L)}$ collects all terms not related to $z^{(L-1)}$. Note that the diagonal matrix $\mathbf{D}^{i,(L-1)}$ implicitly depends on $i$. Then, we merge $\mathbf{A}_{i,:}^{(L-1)}$ with the next linear layer, which is straight forward by plugging in $z^{(L-1)} = \mathbf{W}^{(L-1)} \sigma(z^{(L-2)}) + b^{(L-1)}$:

$$z_i^{(L)} \geq \mathbf{A}_{i,:}^{(L-1)} \mathbf{W}^{(L-1)} \sigma(z^{(L-2)}) + \mathbf{A}_{i,:}^{(L-1)} b^{(L-1)} + \underline{b}_i^{(L-1)}.$$

Then we continue to unfold the next ReLU layer $\sigma(z^{(L-2)})$ using its linear relaxations, and compute a new $\mathbf{A}^{(L-2)} \in \mathbb{R}^{n_L \times n_{L-2}}$ matrix, with $\mathbf{A}_{i,:}^{(L-2)} = \mathbf{A}_{i,:}^{(L-1)} \mathbf{W}^{(L-1)} \mathbf{D}^{i,(L-2)}$ in a similar manner as in (12). Along with the bound propagation process, we need to compute a series of matrices, $\mathbf{A}^{(L-1)}, \cdots, \mathbf{A}^{(0)}$, where $\mathbf{A}_{i,:}^{(l)} = \mathbf{A}_{i,:}^{(l+1)} \mathbf{W}^{(l+1)} \mathbf{D}^{i,(l)} \in \mathbb{R}^{n_L \times n_{(l)}}$, and $\mathbf{A}_{i,:}^{(0)} = \mathbf{A}_{i,:}^{(1)} \mathbf{W}^{(1)} = \mathbf{W}_{i,:}^{(L)} \mathbf{D}^{i,(L-1)} \mathbf{W}^{(L-2)} \mathbf{D}^{i,(L-2)} \mathbf{A}^{(L-2)} \cdots \mathbf{D}^{i,(1)} \mathbf{W}^{(1)}$. At this point, we merged all layers of the network into a linear layer: $z_i^{(L)} \geq \mathbf{A}_{i,:}^{(0)} \boldsymbol{x} + \underline{b}$, where $\underline{b}$ collects all terms not related to $\boldsymbol{x}$. A lower bound for $z_i^{(L)}$ with $\boldsymbol{x}_L \leq \boldsymbol{x} \leq \boldsymbol{x}_U$ can then be easily given as

$$[\underline{m}_{\text{CROWN-IBP}}]_i \equiv \underline{z}_i^{(L)} = \mathbf{A}_{i,:}^{(0)} \boldsymbol{x} + \underline{b} \geq \sum_{k, \mathbf{A}_{i,k}^{(0)} < 0} \mathbf{A}_{i,k}^{(0)} \boldsymbol{x}_{U,k} + \sum_{k, \mathbf{A}_{i,k}^{(0)} > 0} \mathbf{A}_{i,k}^{(0)} \boldsymbol{x}_{L,k} + \underline{b} \quad (13)$$

For ReLU networks, convex adversarial polytope (Wong & Kolter, 2018) uses a very similar bound propagation procedure. CROWN-style bounds allow an adaptive selection of $\alpha_i$ in (10), thus often gives better bounds (e.g., see Table 1). We give details on each term in Appendix L.

**Computational Cost.** Ordinary CROWN (Zhang et al., 2018) and convex adversarial polytope (Wong & Kolter, 2018) use (13) to compute *all intermediate layer's* $\underline{z}_i^{(m)}$ and $\overline{z}_i^{(m)}$ ($m \in [L]$), by considering $\mathbf{W}^{(m)}$ as the final layer of the network. For each layer $m$, we need a different set of $m$ $\mathbf{A}$ matrices, defined as $\mathbf{A}^{m,(l)}, l \in \{m-1, \cdots, 0\}$. This causes three computational issues:

- Unlike the last layer $\mathbf{W}^{(L)}$, an intermediate layer $\mathbf{W}^{(m)}$ typically has a much larger output dimension $n_m \gg n_L$ thus all $\mathbf{A}^{m,(l)} \in \{\mathbf{A}^{m,(m-1)}, \cdots, \mathbf{A}^{m,(0)}\}$ have large dimensions $\mathbb{R}^{n_m \times n_l}$.

- Computation of all $\mathbf{A}^{m,(l)}$ matrices is expensive. Suppose the network has $n$ neurons for all $L-1$ intermediate and input layers and $n_L \ll n$ neurons for the output layer (assuming $L \geq 2$), the time complexity of ordinary CROWN or convex adversarial polytope is $O(\sum_{l=1}^{L-2} ln^3 + (L-1)n_L n^2) = O((L-1)^2 n^3 + (L-1)n_L n^2) = O(Ln^2(Ln + n_L))$. A ordinary forward propagation only takes $O(Ln^2)$ time per example, thus ordinary CROWN does not scale up to large networks for training, due to its quadratic dependency in $L$ and extra $Ln$ times overhead.

- When both $\mathbf{W}^{(l)}$ and $\mathbf{W}^{(l-1)}$ represent convolutional layers with small kernel tensors $\mathbf{K}^{(l)}$ and $\mathbf{K}^{(l-1)}$, there are no efficient GPU operations to form the matrix $\mathbf{W}^{(l)}\mathbf{D}^{(l-1)}\mathbf{W}^{(l-1)}$ using $\mathbf{K}^{(l)}$ and $\mathbf{K}^{(l-1)}$. Existing implementations either unfold at least one of the convolutional kernels to fully connected weights, or use sparse matrices to represent $\mathbf{W}^{(l)}$ and $\mathbf{W}^{(l-1)}$. They suffer from poor hardware efficiency on GPUs.

In CROWN-IBP, we use IBP to obtain bounds of intermediate layers, which takes only twice the regular forward propagate time ($O(Ln^2)$), thus we do not have the first and second issues. The time complexity of the backward bound propagation in CROWN-IBP is $O((L-1)n_L n^2)$, only $n_L$ times slower than forward propagation and significantly more scalable than ordinary CROWN (which is $Ln$ times slower than forward propagation, where typically $n \gg n_L$). The third convolution issue is also not a concern, since we start from the last specification layer $\mathbf{W}^{(L)}$ which is a small fully connected layer. Suppose we need to compute $\mathbf{W}^{(L)}\mathbf{D}^{(L-1)}\mathbf{W}^{(L-1)}$ and $\mathbf{W}^{(L-1)}$ is a convolutional layer with kernel $\mathbf{K}^{(L-1)}$, we can efficiently compute $(\mathbf{W}^{(L-1)\top}(\mathbf{D}^{(L-1)}\mathbf{W}^{(L)\top}))^\top$ on GPUs using the *transposed convolution* operator with kernel $\mathbf{K}^{(L-1)}$, without unfolding any convoluational layers. Conceptually, the backward pass of CROWN-IBP propagates a small specification matrix $\mathbf{W}^{(L)}$ backwards, replacing affine layers with their transposed operators, and activation function layers with a diagonal matrix product. This allows efficient implementation and better scalability.

**Benefits of CROWN-IBP.** Tightness, efficiency and flexibility are unique benefits of CROWN-IBP:

- CROWN-IBP is based on CROWN, a tight linear relaxation based lower bound which can greatly improve the quality of bounds obtained by IBP to guide verifiable training and improve stabability;

- CROWN-IBP avoids the high computational cost of convex relaxation based methods : the time complexity is reduced from $O(Ln^2(Ln + n_L))$ to $O(Ln^2 n_L)$, well suited to problems where the output size $n_L$ is much smaller than input and intermediate layers' sizes; also, there is no quadratic dependency on $L$. Thus, CROWN-IBP is efficient on relatively large networks;

- The objective (9) is strictly more general than IBP and allows the flexibility to exploit the strength from both IBP (good for large $\epsilon$) and convex relaxation based methods (good for small $\epsilon$). We can slowly decrease $\beta$ to 0 during training to avoid the over-regularization problem, yet keeping the initial training of IBP more stable by providing a much tighter bound; we can also keep $\beta = 1$ which helps to outperform convex relaxation based methods in small $\epsilon$ regime (e.g., $\epsilon = 2/255$ on CIFAR-10).

## 4 EXPERIMENTS

**Models and training schedules.** We evaluate CROWN-IBP on three models that are similar to the models used in (Gowal et al., 2018) on MNIST and CIFAR-10 datasets with different $\ell_\infty$ perturbation norms. Here we denote the small, medium and large models in Gowal et al. (2018) as **DM-small**, **DM-medium** and **DM-large**. During training, we first warm up (regular training without robust loss)

for a fixed number of epochs and then increase $\epsilon$ from 0 to $\epsilon_{\text{train}}$ using a ramp-up schedule of $R$ epochs. Similar techniques are also used in many other works (Wong et al., 2018; Wang et al., 2018a; Gowal et al., 2018). For both IBP and CROWN-IBP, a natural cross-entropy (CE) loss with weight $\kappa$ (as in Eq (9)) may be added, and $\kappa$ is scheduled to linearly decrease from $\kappa_{\text{start}}$ to $\kappa_{\text{end}}$ within $R$ ramp-up epochs. Gowal et al. (2018) used $\kappa_{\text{start}} = 1$ and $\kappa_{\text{end}} = 0.5$. To understand the trade-off between verified accuracy and standard (clean) accuracy, we explore two more settings: $\kappa_{\text{start}} = \kappa_{\text{end}} = 0$ (without natural CE loss) and $\kappa_{\text{start}} = 1, \kappa_{\text{end}} = 0$. For $\beta$, a linear schedule during the ramp-up period is used, but we always set $\beta_{\text{start}} = 1$ and $\beta_{\text{end}} = 0$, except that we set $\beta_{\text{start}} = \beta_{\text{end}} = 1$ for CIFAR-10 at $\epsilon = \frac{2}{255}$. Detailed model structures and hyperparameters are in Appendix C. Our training code for IBP and CROWN-IBP, and pre-trained models are publicly available [3].

**Metrics.** Verified error is the percentage of test examples where at least one element in the lower bounds $\underline{\boldsymbol{m}}(\boldsymbol{x}_k, \epsilon)$ is $< 0$. It is an guaranteed upper bound of test error under any $\ell_\infty$ perturbations. We obtain $\underline{\boldsymbol{m}}(\boldsymbol{x}_k, \epsilon)$ using IBP or CROWN-IBP (Eq. 13). We also report standard (clean) errors and errors under 200-step PGD attack. PGD errors are lower bounds of test errors under $\ell_\infty$ perturbations.

**Comparison to IBP.** Table 2 represents the standard, verified and PGD errors under different $\epsilon$ for each dataset with different $\kappa$ settings. We test CROWN-IBP on the same model structures in Table 1 of Gowal et al. (2018). These three models' architectures are presented in Table A in the Appendix. Here we only report the DM-large model structure in as it performs best under all setttings; small and medium models are deferred to Table C in the Appendix. When both $\kappa_{\text{start}} = \kappa_{\text{end}} = 0$, no natural CE loss is added and the model focuses on minimizing verified error, but the lack of natural CE loss may lead to unstable training, especially for IBP; the $\kappa_{\text{start}} = 1, \kappa_{\text{end}} = 0.5$ setting emphasizes on minimizing standard error, usually at the cost of slightly higher verified error rates. $\kappa_{\text{start}} = 1, \kappa_{\text{end}} = 0$ typically achieves the best balance. We can observe that under the same $\kappa$ settings, CROWN-IBP outperforms IBP in *both* standard error and verified error. The benefits of CROWN-IBP is significant especially when model is large and $\epsilon$ is large. We highlight that CROWN-IBP reduces the verified error rate obtained by IBP from 8.21% to 7.02% on MNIST at $\epsilon = 0.3$ and from 55.88% to 46.03% on CIFAR-10 at $\epsilon = 2/255$ (it is the first time that an IBP based method outperforms results from (Wong et al., 2018), and our model also has better standard error). We also note that we are the first to obtain verifiable bound on CIFAR-10 at $\epsilon = 16/255$.

**Trade-off Between Standard Accuracy and Verified Accuracy.** To show the trade-off between standard and verified accuracy, we evaluate DM-large CIFAR-10 model with $\epsilon_{\text{test}} = 8/255$ under different $\kappa$ settings, while keeping all other hyperparameters unchanged. For each $\kappa_{\text{end}} = \{0.5, 0.25, 0\}$, we uniformly choose 11 $\kappa_{\text{start}} \in [1, \kappa_{\text{end}}]$ while keeping all other hyper-parameters unchanged. A larger $\kappa_{\text{start}}$ or $\kappa_{\text{end}}$ tends to produce better standard errors, and we can explicitly control the trade-off between standard accuracy and verified accuracy. In Figure 2 we plot the standard and verified errors of IBP and CROWN-IBP trained models with different $\kappa$ settings. Each cluster on the figure has 11 points, representing 11 different $\kappa_{\text{start}}$ values. Models with lower verified errors tend to have higher standard errors. However, CROWN-IBP clearly outperforms IBP with improvement on both standard and verified accuracy, and

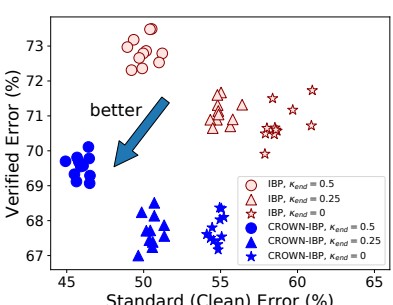

Figure 2: Standard and verified errors of IBP and CROWN-IBP with different $\kappa_{\text{start}}$ and $\kappa_{\text{end}}$ values.

pushes the Pareto front towards the lower left corner, indicating overall better performance. To reach the same verified error of 70%, CROWN-IBP can reduce standard error from roughly 55% to 45%.

**Training Stability.** To discourage hand-tuning on a small set of models and demonstrate the stability of CROWN-IBP over a broader range of models, we evaluate IBP and CROWN-IBP on a variety of small and medium sized model architectures (18 for MNIST and 17 for CIFAR-10), detailed in Appendix D. To evaluate training stability, we compare verified errors under different $\epsilon$ ramp-up schedule length ($R = \{30, 60, 90, 120\}$ on CIFAR-10 and $R = \{10, 15, 30, 60\}$ on MNIST)

---

[3]TensorFlow implementation and pre-trained models: https://github.com/deepmind/interval-bound-propagation/ PyTorch implementation and pre-trained models: https://github.com/huanzhang12/CROWN-IBP

Table 2: The verified, standard (clean) and PGD attack errors for models trained using IBP and CROWN-IBP on MNIST and CIFAR-10. We only present performance on model DM-large here due to limited space (see Table C for a full comparison). CROWN-IBP outperforms IBP under all $\kappa$ settings, and achieves **state-of-the-art** performance on both MNIST and CIFAR datasets for all $\epsilon$.

| Dataset | $\epsilon$ ($\ell_\infty$ norm) | Training Method | $\kappa$ schedules | | Model errors (%) | | | **Best** errors reported in literature (%) | | |
|---|---|---|---|---|---|---|---|---|---|---|
| | | | $\kappa_{\text{start}}$ | $\kappa_{\text{end}}$ | Standard | **Verified** | PGD | Source | Standard | **Verified** |
| MNIST | $\epsilon_{\text{test}} = 0.1$ $\epsilon_{\text{train}} = 0.2$ | IBP | 0 | 0 | 1.13 | 2.89 | 2.24 | Gowal et al. (2018) | 1.06 | 2.92* |
| | | | 1 | 0.5 | 1.08 | 2.75 | 2.02 | Dvijotham et al. (2018b) | 1.2 | 4.44 |
| | | | 1 | 0 | 1.14 | 2.81 | 2.11 | Xiao et al. (2019c) | 1.05 | 4.4 |
| | | CROWN-IBP | 0 | 0 | 1.17 | 2.36 | 1.91 | Wong et al. (2018) | 1.08 | 3.67 |
| | | | 1 | 0.5 | 0.95 | 2.38 | 1.77 | Mirman et al. (2018) | 1.0 | 3.4 |
| | | | 1 | 0 | 1.17 | **2.24** | 1.81 | | | |
| | $\epsilon_{\text{test}} = 0.2$ $\epsilon_{\text{train}} = 0.4$ | IBP | 0 | 0 | 3.45 | 6.46 | 6.00 | Gowal et al. (2018) | 1.66 | 4.53* |
| | | | 1 | 0.5 | 2.12 | 4.75 | 4.24 | Xiao et al. (2019c) | 1.9 | 10.21 |
| | | | 1 | 0 | 2.74 | 5.46 | 4.89 | | | |
| | | CROWN-IBP | 0 | 0 | 2.84 | 5.15 | 4.90 | | | |
| | | | 1 | 0.5 | 1.82 | **4.13** | 3.81 | | | |
| | | | 1 | 0 | 2.17 | 4.31 | 3.99 | | | |
| | $\epsilon_{\text{test}} = 0.3$ $\epsilon_{\text{train}} = 0.4$ | IBP | 0 | 0 | 3.45 | 9.76 | 8.42 | Gowal et al. (2018) | 1.66 | 8.21* |
| | | | 1 | 0.50 | 2.12 | 8.47 | 6.78 | Wong et al. (2018) | 14.87 | 43.1 |
| | | | 1 | 0 | 2.74 | 8.73 | 7.37 | Xiao et al. (2019c) | 2.67 | 19.32 |
| | | CROWN-IBP | 0 | 0 | 2.84 | 7.65 | 6.90 | | | |
| | | | 1 | 0.5 | 1.82 | **7.02** | 6.05 | | | |
| | | | 1 | 0 | 2.17 | 7.03 | 6.12 | | | |
| | $\epsilon_{\text{test}} = 0.4$ $\epsilon_{\text{train}} = 0.4$ | IBP | 0 | 0 | 3.45 | 16.19 | 12.73 | Gowal et al. (2018) | 1.66 | 15.01* |
| | | | 1 | 0.5 | 2.12 | 15.37 | 11.05 | | | |
| | | | 1 | 0 | 2.74 | 14.80 | 11.14 | | | |
| | | CROWN-IBP | 0 | 0 | 2.84 | 12.74 | 10.39 | | | |
| | | | 1 | 0.5 | 1.82 | 12.59 | 9.58 | | | |
| | | | 1 | 0 | 2.17 | **12.06** | 9.47 | | | |
| CIFAR-10 | $\epsilon_{\text{test}} = \frac{2}{255}$ § $\epsilon_{\text{train}} = \frac{2.2}{255}$ ‡ | IBP | 0 | 0 | 38.54 | 55.21 | 49.72 | Gowal et al. (2018) | 29.84 | 55.88* |
| | | | 1 | 0.5 | 33.77 | 58.48 | 50.54 | Mirman et al. (2018) | 38.0 | 47.8 |
| | | | 1 | 0 | 39.22 | 55.19 | 50.40 | Wong et al. (2018) | 31.72 | 46.11 |
| | | CROWN-IBP | 0 | 0 | 28.48 | **46.03** | 40.28 | Xiao et al. (2019c) | 38.88 | 54.07 |
| | | | 1 | 0.5 | 26.19 | 50.53 | 40.24 | | | |
| | | | 1 | 0 | 28.91 | 46.43 | 40.27 | | | |
| | $\epsilon_{\text{test}} = \frac{8}{255}$ $\epsilon_{\text{train}} = \frac{8.8}{255}$ ‡ | IBP | 0 | 0 | 59.41 | 71.22 | 68.96 | Gowal et al. (2018) | 50.51 | (68.44)† |
| | | | 1 | 0.5 | 49.01 | 72.68 | 68.14 | Dvijotham et al. (2018b) | 51.36 | 73.33 |
| | | | 1 | 0 | 58.43 | 70.81 | 68.73 | Xiao et al. (2019c) | 59.55 | 79.73 |
| | | CROWN-IBP | 0 | 0 | 54.02 | **66.94** | 65.42 | Wong et al. (2018) | 71.33 | 78.22 |
| | | | 1 | 0.5 | 45.47 | 69.55 | 65.74 | Mirman et al. (2019) | 59.8 | 76.8 |
| | | | 1 | 0 | 55.27 | 67.76 | 65.71 | | | |
| | $\epsilon_{\text{test}} = \frac{16}{255}$ $\epsilon_{\text{train}} = \frac{17.6}{255}$ ‡ | IBP | 0 | 0 | 68.97 | 78.12 | 76.66 | None, but our best verified test error (76.80%) and standard test error (66.06%) are both better than Wong et al. (2018) at $\epsilon = \frac{8}{255}$, despite our $\epsilon$ being twice larger. | | |
| | | | 1 | 0.5 | 59.46 | 80.85 | 76.97 | | | |
| | | | 1 | 0 | 68.88 | 78.91 | 76.95 | | | |
| | | CROWN-IBP | 0 | 0 | 67.17 | 77.27 | 75.76 | | | |
| | | | 1 | 0.5 | 56.73 | 78.20 | 74.87 | | | |
| | | | 1 | 0 | 66.06 | **76.80** | 75.23 | | | |

* Verified errors reported in Table 4 of Gowal et al. (2018) are evaluated using mixed integer programming (MIP) and linear programming (LP), which are strictly smaller than IBP verified errors but computationally expensive. For a fair comparison, we use the IBP verified errors reported in their Table 3.
† According to direct communications with Gowal et al. (2018), achieving the 68.44% IBP verified error requires to adding an extra PGD adversarial training loss. Without adding PGD, the verified error is 72.91% (LP/MIP verified) or 73.52% (IBP verified). Our result should be compared to 73.52%.
‡ Although not explicitly mentioned, the CIFAR-10 models in (Gowal et al., 2018) are trained using $\epsilon_{\text{train}} = 1.1\epsilon_{\text{test}}$. We thus follow their settings.
§ We use $\beta_{\text{start}} = \beta_{\text{end}} = 1$ for this setting, and thus CROWN-IBP bound ($\beta = 1$) is used to evaluate the verified error.

and different $\kappa$ settings. Instead of reporting just the best model, we compare the best, worst and median verified errors over all models. Our results are presented in Figure 3: (a) is for MNIST with $\epsilon = 0.3$; (c),(d) are for CIFAR with $\epsilon = 8/255$. We can observe that CROWN-IBP achieves better performance consistently under different schedule length. In addition, IBP with $\kappa = 0$ cannot stably converge on all models when $\epsilon$ schedule is short; under other $\kappa$ settings, CROWN-IBP always performs better. We conduct additional training stability experiments on MNIST and CIFAR-10 dataset under other model and $\epsilon$ settings and the observations are similar (see Appendix H).

## 5 CONCLUSIONS

We propose a new certified defense method, CROWN-IBP, by combining the fast interval bound propagation (IBP) bound and a tight linear relaxation based bound, CROWN. Our method enjoys high computational efficiency provided by IBP while facilitating the tight CROWN bound to stabilize training under the robust optimization framework, and provides the flexibility to trade-off between the two. Our experiments show that CROWN-IBP consistently outperforms other IBP baselines in both standard errors and verified errors and achieves state-of-the-art verified test errors for $\ell_\infty$ robustness.

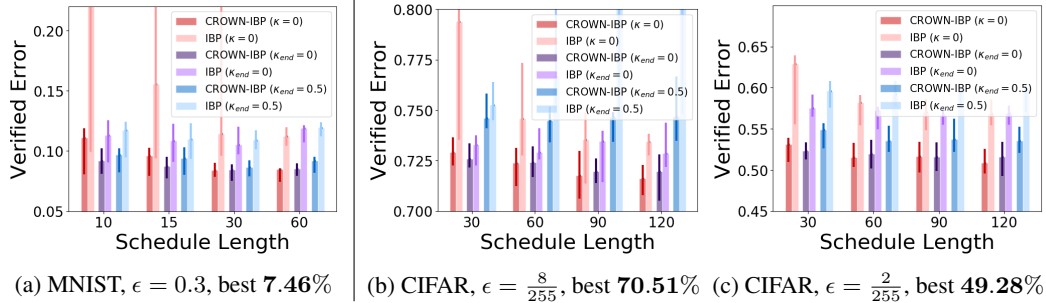

(a) MNIST, $\epsilon = 0.3$, best **7.46**% | (b) CIFAR, $\epsilon = \frac{8}{255}$, best **70.51**% (c) CIFAR, $\epsilon = \frac{2}{255}$, best **49.28**%

Figure 3: Verified error vs. schedule length on 8 medium MNIST models and 8 medium CIFAR-10 models. The solid bars show median values of verified errors. $\kappa_{\text{start}} = 1.0$ except for the $\kappa = 0$ setting. The upper and lower ends of an error bar are the worst and best verified error, respectively. For each schedule length, three color groups represent three different $\kappa$ settings.

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

## A   IBP as a Simple Augmented Network

Despite achieving great success, it is still an open question why IBP based methods significantly outperform convex relaxation based methods, despite the fact that convex relaxations usually provide significantly tighter bounds. We conjecture that IBP performs better because the bound propagation process can be viewed as a *ReLU network with the same depth* as the original network, and the IBP training process is effectively training this equivalent network for standard accuracy, as explained below.

Given a fixed neural network (NN) $f(\boldsymbol{x})$, IBP gives a very loose estimation of the output range of $f(\boldsymbol{x})$. However, during training, since the weights of this NN can be updated, we can equivalently view IBP as an augmented neural network, which we denote as an IBP-NN (Figure A). Unlike a usual network which takes an input $\boldsymbol{x}_k$ with label $y_k$, IBP-NN takes two points $\boldsymbol{x}_L = \boldsymbol{x}_k - \epsilon$ and $\boldsymbol{x}_U = \boldsymbol{x}_k + \epsilon$ as inputs (where $\boldsymbol{x}_L \leq \boldsymbol{x} \leq \boldsymbol{x}_U$, element-wisely). The bound propagation process can be equivalently seen as forward propagation in a specially structured neural network, as shown in Figure A. After the last specification layer $C$ (typically merged into $\mathbf{W}^{(L)}$), we can obtain $\underline{\boldsymbol{m}}(\boldsymbol{x}_k, \epsilon)$. Then, $-\underline{\boldsymbol{m}}(\boldsymbol{x}_k, \epsilon)$ is sent to softmax layer for prediction. Importantly, since $[\underline{\boldsymbol{m}}(\boldsymbol{x}_k, \epsilon)]_{y_k} = 0$ (as the $y_k$-th row in $C$ is always 0), the top-1 prediction of the augmented IBP network is $y_k$ if and only if all other elements of $\underline{\boldsymbol{m}}(\boldsymbol{x}_k, \epsilon)$ are positive, i.e., the original network will predict correctly for all $\boldsymbol{x}_L \leq \boldsymbol{x} \leq \boldsymbol{x}_U$. When we train the augmented IBP network with ordinary cross-entropy loss and desire it to predict correctly on an input $\boldsymbol{x}_k$, we are implicitly doing robust optimization (Eq. (2)).

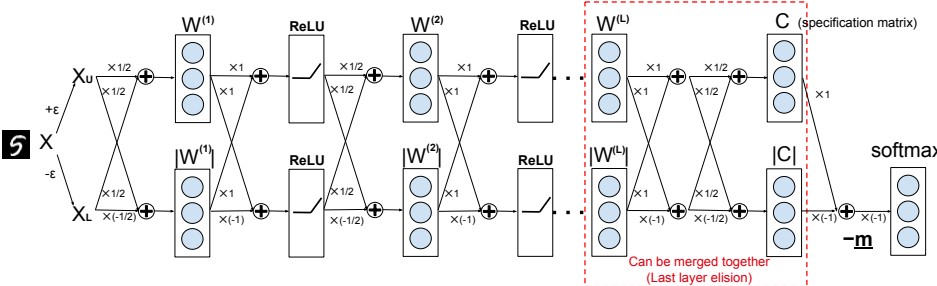

Figure A: Interval Bound Propagation viewed as training an *augmented neural network* (IBP-NN). The inputs of IBP-NN are two images $\boldsymbol{x}_k + \epsilon$ and $\boldsymbol{x}_k - \epsilon$. The output of IBP-NN is a vector of lower bounds of margins (denoted as $\underline{\boldsymbol{m}}$) between ground-truth class and all classes (including the ground-truth class itself) for all $\boldsymbol{x}_k - \epsilon \leq \boldsymbol{x} \leq \boldsymbol{x}_k + \epsilon$. This vector $\underline{\boldsymbol{m}}$ is negated and sent into a regular softmax function to get model prediction. The top-1 prediction of softmax is correct if and only if all margins between the ground-truth class and other classes (except the ground truth class) are positive, i.e., the model is verifiably robust. Thus, an IBP-NN with low standard error guarantees low verified error on the original network.

The simplicity of IBP-NN may help a gradient based optimizer to find better solutions. On the other hand, while the computation of convex relaxation based bounds can also be cast as an equivalent network (e.g., the "dual network" in Wong & Kolter (2018)), its construction is significantly more complex, and sometimes requires non-differentiable indicator functions (the sets $\mathcal{I}^+$, $\mathcal{I}^-$ and $\mathcal{I}$ in Wong & Kolter (2018)). As a consequence, it can be challenging for the optimizer to find a good solution, and the optimizer tends to making the bounds tighter naively by reducing the norm of weight matrices and over-regularizing the network, as demonstrated in Figure 1.

## B   Tightness comparison between IBP and CROWN-IBP

Both IBP and CROWN-IBP produce lower bounds $\underline{\boldsymbol{m}}(\boldsymbol{x}, \epsilon)$, and a larger lower bound has better quality. To measure the relative tightness of the two bounds, we take the average of all bounds of training examples:

$$\underset{(\boldsymbol{x}, y) \in \mathcal{X}}{E} \frac{1}{n_L} \mathbf{1}^{\top} (\underline{\boldsymbol{m}}_{\text{CROWN-IBP}}(\boldsymbol{x}, \epsilon) - \underline{\boldsymbol{m}}_{\text{IBP}}(\boldsymbol{x}, \epsilon))$$

A positive value indicates that CROWN-IBP is tighter than IBP. In Figure B we plot this averaged bound differences during $\epsilon$ schedule for one MNIST model and one CIFAR-10 model. We can observe that during the early phase of training when the $\epsilon$ schedule just starts, CROWN-IBP produces significantly better bounds than IBP. A tighter lower bound $\underline{m}(x, \epsilon)$ gives a tighter upper bound for $\max_{\delta \in S} L(x + \delta; y; \theta)$, making the minimax optimization problem (2) more effective to solve. When the training schedule proceeds, the model gradually learns how to make IBP bounds tighter and eventually the difference between the two bounds become close to 0.

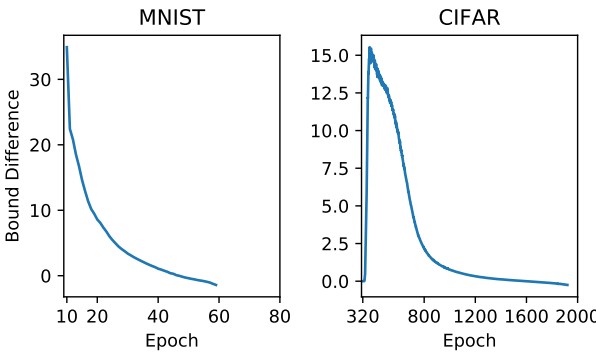

Figure B: Bound differences between IBP and CROWN-IBP for DM-large models during training. The bound difference is only computed during the $\epsilon$ schedule (epoch 10 to 60 for MNIST, and 320 to 1920 for CIFAR-10), as we don't compute CROWN-IBP bounds in warmup period and after $\epsilon$ schedule.

**Why CROWN-IBP stabilizes IBP training?**   When taking a randomly initialized network or a naturally trained network, IBP bounds are very loose. But in Table 1, we show that a network trained using IBP can eventually obtain quite tight IBP bounds and high verified accuracy; the network can adapt to IBP bounds and learn a specific set of weights to make IBP tight and also correctly classify examples. However, since the training has to start from weights that produce loose bounds for IBP, the beginning phase of IBP training can be challenging and is vitally important.

We observe that IBP training can have a large performance variance across models and initializations. Also IBP is more sensitive to hyper-parameter like $\kappa$ or schedule length; in Figure 3, many IBP models converge sub-optimally (large worst/median verified error). The reason for instability is that during the beginning phase of training, the loose bounds produced by IBP make the robust loss (9) ineffective, and it is challenging for the optimizer to reduce this loss and find a set of good weights that produce tight IBP verified bounds in the end.

Conversely, if our bounds are much tighter at the beginning, the robust loss (9) always remains in a reasonable range during training, and the network can gradually learn to find a good set of weights that make IBP bounds increasingly tighter (this is obvious in Figure B). Initially, tighter bounds can be provided by a convex relaxation based method like CROWN, and they are gradually replaced by IBP bounds (using $\beta_{\text{start}} = 1, \beta_{\text{end}} = 0$), eventually leading to a model with learned tight IBP bounds in the end.

## C   MODELS AND HYPERPARAMETERS FOR COMPARISON TO IBP

The goal of these experiments is to reproduce the performance reported in (Gowal et al., 2018) and demonstrate the advantage of CROWN-IBP under the same experimental settings. Specifically, to reproduce the IBP results, for CIFAR-10 we train using a large batch size and long training schedule on TPUs (we can also replicate these results on multi-GPUs using a reasonable amount of training time; see Section F). Also, for this set of experiments we use the same code base as in Gowal et al. (2018). For model performance on a comprehensive set of small and medium sized models trained on a single GPU, please see Table D in Section F, as well as the training stability experiments in Section 4 and Section H.

The models structures (DM-small, DM-medium and DM-large) used in Table C and Table 2 are listed in Table A. These three model structures are the same as in Gowal et al. (2018). Training hyperparameters are detailed below:

- For MNIST IBP baseline results, we follow exact the same set of hyperparameters as in (Gowal et al., 2018). We train 100 epochs (60K steps) with a batch size of 100, and use a warm-up and ramp-up duration of 2K and 10K steps. Learning rate for Adam optimizer is set to $1 \times 10^{-3}$ and decayed by 10X at steps 15K and 25K. Our IBP results match their reported numbers. Note that we always use IBP verified errors rather than MIP verified errors. We use the same schedule for CROWN-IBP with $\epsilon_{\text{train}} = 0.2$ ($\epsilon_{\text{test}} = 0.1$) in Table C and Table 2. For $\epsilon_{\text{train}} = 0.4$, this schedule can obtain verified error rates 4.22%, 7.01% and 12.84% at $\epsilon_{\text{test}} = \{0.2, 0.3, 0.4\}$ using the DM-Large model, respectively.

- For MNIST CROWN-IBP with $\epsilon_{\text{train}} = 0.4$ in Table C and Table 2, we train 200 epochs with a batch size of 256. We use Adam optimizer and set learning rate to $5 \times 10^{-4}$. We warm up with 10 epochs' regular training, and gradually ramp up $\epsilon$ from 0 to $\epsilon_{\text{train}}$ in 50 epochs. We reduce the learning rate by 10X at epoch 130 and 190. Using this schedule, IBP's performance becomes worse (by about 1-2% in all settings), but this schedule improves verified error for CROWN-IBP at $\epsilon_{\text{test}} = 0.4$ from 12.84% to to 12.06% and does do affect verified errors at other $\epsilon_{\text{test}}$ levels.

- For CIFAR-10, we follow the setting in Gowal et al. (2018) and train 3200 epochs on 32 TPU cores. We use a batch size of 1024, and a learning rate of $5 \times 10^{-4}$. We warm up for 320 epochs, and ramp-up $\epsilon$ for 1600 epochs. Learning rate is reduced by 10X at epoch 2600 and 3040. We use random horizontal flips and random crops as data augmentation, and normalize images according to per-channel statistics. Note that this schedule is slightly different from the schedule used in (Gowal et al., 2018); we use a smaller batch size due to TPU memory constraints (we used TPUv2 which has half memory capacity as TPUv3 used in (Gowal et al., 2018)), and also we decay learning rates later. We found that this schedule improves both IBP baseline performance and CROWN-IBP performance by around 1%; for example, at $\epsilon = 8/255$, this improved schedule can reduce verified error from 73.52% to 72.68% for IBP baseline ($\kappa_{\text{start}} = 1.0$, $\kappa_{\text{end}} = 0.5$) using the DM-Large model.

**Hyperparameter $\kappa$ and $\beta$.** We use a linear schedule for both hyperparameters, decreasing $\kappa$ from $\kappa_{\text{start}}$ to $\kappa_{\text{end}}$ while increasing $\beta$ from $\beta_{\text{start}}$ to $\beta_{\text{end}}$. The schedule length is set to the same length as the $\epsilon$ schedule.

In both IBP and CROWN-IBP, a hyperparameter $\kappa$ is used to trade-off between clean accuracy and verified accuracy. Figure 2 shows that $\kappa_{\text{end}}$ can significantly affect the trade-off, while $\kappa_{\text{start}}$ has minor impacts compared to $\kappa_{\text{end}}$. In general, we recommend $\kappa_{\text{start}} = 1$ and $\kappa_{\text{end}} = 0$ as a safe starting point, and we can adjust $\kappa_{\text{end}}$ to a larger value if a better standard accuracy is desired. The setting $\kappa_{\text{start}} = \kappa_{\text{end}} = 0$ (pure minimax optimization) can be challenging for IBP as there is no natural loss as a stabilizer; under this setting CROWN-IBP usually produces a model with good (sometimes best) verified accuracy but noticeably worse standard accuracy (on CIFAR-10 $\epsilon = \frac{8}{255}$ the difference can be as large as 10%), so this setting is only recommended when a model with best verified accuracy is desired at a cost of noticeably reduced standard accuracy.

Compared to IBP, CROWN-IBP adds one additional hyperparameter, $\beta$. $\beta$ has a clear meaning: balancing between the convex relaxation based bounds and the IBP bounds. $\beta_{\text{start}}$ is always set to 1 as we want to use CROWN-IBP to obtain tighter bounds to stabilize the early phase of training when IBP bounds are very loose; $\beta_{\text{end}}$ determines if we want to use a convex relaxation based bound ($\beta_{\text{end}} = 1$) or IBP based bound ($\beta_{\text{end}} = 0$) after the $\epsilon$ schedule. Thus, we set $\beta_{\text{end}} = 1$ for the case where convex relaxation based method (Wong et al., 2018) can outperform IBP (e.g., CIFAR-10 $\epsilon = 2/255$, and $\beta_{\text{end}} = 0$ for the case where IBP outperforms convex relaxation based bounds. We do not tune or grid-search this hyperparameter.

| DM-Small | DM-Medium | DM-Large |
|---|---|---|
| CONV 16 4×4+2 | CONV 32 3×3+1 | CONV 64 3×3+1 |
| CONV 32 4×4+1 | CONV 32 4×4+2 | CONV 64 3×3+1 |
| FC 100 | CONV 64 3×3+1 | CONV 128 3×3+2 |
|  | CONV 64 4×4+2 | CONV 128 3×3+1 |
|  | FC 512 | CONV 128 3×3+1 |
|  | FC 512 | FC 512 |

Table A: Model structures from Gowal et al. (2018). "CONV $k$ $w$×$h$+$s$" represents a 2D convolutional layer with $k$ filters of size $w$×$h$ using a stride of $s$ in both dimensions. "FC n" = fully connected layer with $n$ outputs. Last fully connected layer is omitted. All networks use ReLU activation functions.

## D  HYPERPARAMETERS AND MODEL STRUCTURES FOR TRAINING STABILITY EXPERIMENTS

In all our training stability experiments, we use a large number of relatively small models and train them on a single GPU. These small models cannot achieve state-of-the-art performance but they can be trained quickly and cheaply, allowing us to explore training stability over a variety of settings, and report min, median and max statistics. We use the following hyperparameters:

- For MNIST, we train 100 epochs with batch size 256. We use Adam optimizer and the learning rate is $5 \times 10^{-4}$. The first epoch is standard training for warming up. We gradually increase $\epsilon$ linearly per batch in our training process with a $\epsilon$ schedule length of 60. We reduce the learning rate by 50% every 10 epochs after $\epsilon$ schedule ends. No data augmentation technique is used and the whole $28 \times 28$ images are used (normalized to 0 - 1 range).

- For CIFAR, we train 200 epoch with batch size 128. We use Adam optimizer and the learning rate is 0.1%. The first 10 epochs are standard training for warming up. We gradually increase $\epsilon$ linearly per batch in our training process with a $\epsilon$ schedule length of 120. We reduce the learning rate by 50% every 10 epochs after $\epsilon$ schedule ends. We use random horizontal flips and random crops as data augmentation. The three channels are normalized with mean (0.4914, 0.4822, 0.4465) and standard deviation (0.2023, 0.1914, 0.2010). These numbers are per-channel statistics from the training set used in (Gowal et al., 2018).

All verified error numbers are evaluated on the test set using IBP, since the networks are trained using IBP ($\beta = 0$ after $\epsilon$ reaches the target $\epsilon_{\text{train}}$), except for CIFAR $\epsilon = \frac{2}{255}$ where we set $\beta = 1$ to compute the CROWN-IBP verified error.

Table B gives the 18 model structures used in our training stability experiments. These model structures are designed by us and are not used in Gowal et al. (2018). Most CIFAR-10 models share the same structures as MNIST models (unless noted on the table) except that their input dimensions are different. Model A is too small for CIFAR-10 thus we remove it for CIFAR-10 experiments. Models A - J are the "small models" reported in Figure 3. Models K - T are the "medium models" reported in Figure 3. For results in Table 1, we use a small model (model structure B) for all three datasets. These MNIST, CIFAR-10 models can be trained on a single NVIDIA RTX 2080 Ti GPU within a few hours each.

## E  OMITTED RESULTS ON DM-SMALL AND DM-MEDIUM MODELS

In Table 2 we report results from the best DM-Large model. Table C presents the verified, standard (clean) and PGD attack errors for all three model structures used in (Gowal et al., 2018) (DM-Small, DM-Medium and DM-Large) trained on MNIST and CIFAR-10 datasets. We evaluate IBP and CROWN-IBP under the same three $\kappa$ settings as in Table 2. We use hyperparameters detailed in Section C to train these models. We can see that given any model structure and any $\kappa$ setting, CROWN-IBP consistently outperforms IBP.

| Name | Model Structure (all models have a last FC 10 layer, which are omitted) |
|---|---|
| A (MNIST Only) | Conv 4 $4 \times 4$+2, Conv 8 $4 \times 4$+2, FC 128 |
| B | Conv 8 $4 \times 4$+2, Conv 16 $4 \times 4$+2, FC 256 |
| C | Conv 4 $3 \times 3$+1, Conv 8 $3 \times 3$+1, Conv 8 $4 \times 4$+4, FC 64 |
| D | Conv 8 $3 \times 3$+1, Conv 16 $3 \times 3$+1, Conv 16 $4 \times 4$+4, FC 128 |
| E | Conv 4 $5 \times 5$+1, Conv 8 $5 \times 5$+1, Conv 8 $5 \times 5$+4, FC 64 |
| F | Conv 8 $5 \times 5$+1, Conv 16 $5 \times 5$+1, Conv 16 $5 \times 5$+4, FC 128 |
| G | Conv 4 $3 \times 3$+1, Conv 4 $4 \times 4$+2, Conv 8 $3 \times 3$+1, Conv 8 $4 \times 4$+2, FC 256, FC 256 |
| H | Conv 8 $3 \times 3$+1, Conv 8 $4 \times 4$+2, Conv 16 $3 \times 3$+1, Conv 16 $4 \times 4$+2, FC 256, FC 256 |
| I | Conv 4 $3 \times 3$+1, Conv 4 $4 \times 4$+2, Conv 8 $3 \times 3$+1, Conv 8 $4 \times 4$+2, FC 512, FC 512 |
| J | Conv 8 $3 \times 3$+1, Conv 8 $4 \times 4$+2, Conv 16 $3 \times 3$+1, Conv 16 $4 \times 4$+2, FC 512, FC 512 |
| K | Conv 16 $3 \times 3$+1, Conv 16 $4 \times 4$+2, Conv 32 $3 \times 3$+1, Conv 32 $4 \times 4$+2, FC 256, FC 256 |
| L | Conv 16 $3 \times 3$+1, Conv 16 $4 \times 4$+2, Conv 32 $3 \times 3$+1, Conv 32 $4 \times 4$+2, FC 512, FC 512 |
| M | Conv 32 $3 \times 3$+1, Conv 32 $4 \times 4$+2, Conv 64 $3 \times 3$+1, Conv 64 $4 \times 4$+2, FC 512, FC 512 |
| N | Conv 64 $3 \times 3$+1, Conv 64 $4 \times 4$+2, Conv 128 $3 \times 3$+1, Conv 128 $4 \times 4$+2, FC 512, FC 512 |
| O(MNIST Only) | Conv 64 $5 \times 5$+1, Conv 128 $5 \times 5$+1, Conv 128 $4 \times 4$+4, FC 512 |
| P(MNIST Only) | Conv 32 $5 \times 5$+1, Conv 64 $5 \times 5$+1, Conv 64 $4 \times 4$+4, FC 512 |
| Q | Conv 16 $5 \times 5$+1, Conv 32 $5 \times 5$+1, Conv 32 $5 \times 5$+4, FC 512 |
| R | Conv 32 $3 \times 3$+1, Conv 64 $3 \times 3$+1, Conv 64 $3 \times 3$+4, FC 512 |
| S(CIFAR-10 Only) | Conv 32 $4 \times 4$+2, Conv 64 $4 \times 4$+2, FC 128 |
| T(CIFAR-10 Only) | Conv 64 $4 \times 4$+2, Conv 128 $4 \times 4$+2, FC 256 |

Table B: Model structures used in our training stability experiments. We use ReLU activations for all models. We omit the last fully connected layer as its output dimension is always 10. In the table, "Conv $k$ $w \times w + s$" represents to a 2D convolutional layer with $k$ filters of size $w \times w$ and a stride of $s$. Model A - J are referred to as "small models" and model K to T are referred to as "medium models".

## F ADDITIONAL EXPERIMENTS ON SMALLER MODELS USING A SINGLE GPU

In this section we present additional experiments on a variety of smaller MNIST and CIFAR-10 models which can be trained on a single GPU. The purpose of this experiment is to compare model performance statistics (min, median and max) *on a wide range of models*, rather than a few hand selected models. The model structures used in these experiments are detailed in Table B. In Table D, we present the best, median and worst verified and standard (clean) test errors for models trained on MNIST and CIFAR-10 using IBP and CROWN-IBP. Although these small models cannot achieve state-of-the-art performance, CROWN-IBP's best, median and worst verified errors among all model structures consistently outperform those of IBP. Especially, in many situations the worst case verified error improves *significantly* using CROWN-IBP, because IBP training is not stable on some of the models.

It is worth noting that in this set of experiments we explore a different $\epsilon$ setting: $\epsilon_{\text{train}} = \epsilon_{\text{test}}$. We found that both IBP and CROWN-IBP tend to overfit to training dataset on MNIST with small $\epsilon$, thus verified errors are not as good as presented in Table C. This overfitting issue can be alleviated by using $\epsilon_{\text{train}} > \epsilon_{\text{test}}$ (as used in Table 2 and Table C), or using an explicit $\ell_1$ regularization, which will be discussed in detail in Section I.

Table C: The verified, standard (clean) and PGD attack errors for 3 models (DM-small, DM-medium, DM-large) trained on MNIST and CIFAR test sets. We evaluate IBP and CROWN-IBP under different $\kappa$ schedules. CROWN-IBP outperforms IBP under the same $\kappa$ setting, and also achieves state-of-the-art results for $\ell_\infty$ robustness on both MNIST and CIFAR datasets for all $\epsilon$.

| Dataset | $\epsilon$ ($\ell_\infty$ norm) | Training Method | $\kappa$ schedules | | **DM-small** model's err. (%) | | | **DM-medium** model's err. (%) | | | **DM-large** model's err. (%) | | |
|---|---|---|---|---|---|---|---|---|---|---|---|---|---|
| | | | $\kappa_{\text{start}}$ | $\kappa_{\text{end}}$ | Standard | **Verified** | PGD | Standard | **Verified** | PGD | Standard | **Verified** | PGD |
| MNIST | $\epsilon_{\text{test}} = 0.1$ $\epsilon_{\text{train}} = 0.2$ | IBP | 0 | 0 | 1.92 | 4.16 | 3.88 | 1.53 | 3.26 | 2.82 | 1.13 | 2.89 | 2.24 |
| | | | 1 | 0.5 | 1.68 | 3.60 | 3.34 | 1.46 | 3.20 | 2.57 | 1.08 | 2.75 | 2.02 |
| | | | 1 | 0 | 2.14 | 4.24 | 3.94 | 1.48 | 3.21 | 2.77 | 1.14 | 2.81 | 2.11 |
| | | CROWN-IBP | 0 | 0 | 1.90 | 3.50 | 3.21 | 1.44 | 2.77 | 2.37 | 1.17 | 2.36 | 1.91 |
| | | | 1 | 0.5 | 1.60 | 3.51 | 3.19 | 1.14 | **2.64** | 2.23 | 0.95 | 2.38 | 1.77 |
| | | | 1 | 0 | 1.67 | **3.44** | 3.09 | 1.34 | 2.76 | 2.39 | 1.17 | **2.24** | 1.81 |
| | $\epsilon_{\text{test}} = 0.2$ $\epsilon_{\text{train}} = 0.4$ | IBP | 0 | 0 | 5.08 | 9.80 | 9.36 | 3.68 | 7.38 | 6.77 | 3.45 | 6.46 | 6.00 |
| | | | 1 | 0.5 | 3.83 | 8.64 | 8.06 | 2.55 | 5.84 | 5.33 | 2.12 | 4.75 | 4.24 |
| | | | 1 | 0 | 6.25 | 11.32 | 10.84 | 3.89 | 7.21 | 6.68 | 2.74 | 5.46 | 4.89 |
| | | CROWN-IBP | 0 | 0 | 3.78 | 6.61 | 6.40 | 3.84 | 6.65 | 6.42 | 2.84 | 5.15 | 4.90 |
| | | | 1 | 0.5 | 2.96 | **6.11** | 5.74 | 2.37 | **5.35** | 4.90 | 1.82 | **4.13** | 3.81 |
| | | | 1 | 0 | 3.55 | 6.29 | 6.13 | 3.16 | 5.82 | 5.44 | 2.17 | 4.31 | 3.99 |
| | $\epsilon_{\text{test}} = 0.3$ $\epsilon_{\text{train}} = 0.4$ | IBP | 0 | 0 | 5.08 | 14.42 | 13.30 | 3.68 | 10.97 | 9.66 | 3.45 | 9.76 | 8.42 |
| | | | 1 | 0.50 | 3.83 | 13.99 | 12.25 | 2.55 | 9.51 | 7.87 | 2.12 | 8.47 | 6.78 |
| | | | 1 | 0 | 6.25 | 16.51 | 15.07 | 3.89 | 10.4 | 9.17 | 2.74 | 8.73 | 7.37 |
| | | CROWN-IBP | 0 | 0 | 3.78 | 9.60 | 8.90 | 3.84 | 9.25 | 8.57 | 2.84 | 7.65 | 6.90 |
| | | | 1 | 0.5 | 2.96 | 9.44 | 8.26 | 2.37 | **8.54** | 7.74 | 1.82 | **7.02** | 6.05 |
| | | | 1 | 0 | 3.55 | **9.40** | 8.50 | 3.16 | 8.62 | 7.65 | 2.17 | 7.03 | 6.12 |
| | $\epsilon_{\text{test}} = 0.4$ $\epsilon_{\text{train}} = 0.4$ | IBP | 0 | 0 | 5.08 | 23.40 | 20.15 | 3.68 | 18.34 | 14.75 | 3.45 | 16.19 | 12.73 |
| | | | 1 | 0.5 | 3.83 | 24.16 | 19.97 | 2.55 | 16.82 | 12.83 | 2.12 | 15.37 | 11.05 |
| | | | 1 | 0 | 6.25 | 26.81 | 22.78 | 3.89 | 16.99 | 13.81 | 2.74 | 14.80 | 11.14 |
| | | CROWN-IBP | 0 | 0 | 3.78 | **15.21** | 13.34 | 3.84 | 14.58 | 12.69 | 2.84 | 12.74 | 10.39 |
| | | | 1 | 0.5 | 2.96 | 16.04 | 12.91 | 2.37 | 14.97 | 12.47 | 1.82 | 12.59 | 9.58 |
| | | | 1 | 0 | 3.55 | 15.55 | 13.11 | 3.16 | **14.19** | 11.31 | 2.17 | **12.06** | 9.47 |
| CIFAR-10 | $\epsilon_{\text{test}} = \frac{2}{255}$ [4] $\epsilon_{\text{train}} = \frac{2.2}{255}$ [3] | IBP | 0 | 0 | 44.66 | 56.38 | 54.15 | 39.12 | 53.86 | 49.77 | 38.54 | 55.21 | 49.72 |
| | | | 1 | 0.5 | 38.90 | 57.94 | 53.64 | 34.19 | 56.24 | 49.63 | 33.77 | 58.48 | 50.54 |
| | | | 1 | 0 | 44.08 | 56.32 | 54.16 | 39.30 | 53.68 | 49.74 | 39.22 | 55.19 | 50.40 |
| | | CROWN-IBP | 0 | 0 | 39.43 | 53.93 | 49.16 | 32.78 | **49.57** | 44.22 | 28.48 | **46.03** | 40.28 |
| | | | 1 | 0.5 | 34.08 | 54.28 | 51.17 | 28.63 | 51.39 | 42.43 | 26.19 | 50.53 | 40.24 |
| | | | 1 | 0 | 38.15 | **52.57** | 50.35 | 33.17 | 49.82 | 44.64 | 28.91 | 46.43 | 40.27 |
| | $\epsilon_{\text{test}} = \frac{8}{255}$ $\epsilon_{\text{train}} = \frac{8.8}{255}$ [3] | IBP | 0 | 0 | 61.91 | 73.12 | 71.75 | 61.46 | 71.98 | 70.07 | 59.41 | 71.22 | 68.96 |
| | | | 1 | 0.5 | 54.01 | 73.04 | 70.54 | 50.33 | 73.58 | 69.57 | 49.01 | 72.68 | 68.14 |
| | | | 1 | 0 | 62.66 | 72.25 | 70.98 | 61.61 | 72.60 | 70.57 | 58.43 | 70.81 | 68.73 |
| | | CROWN-IBP | 0 | 0 | 59.94 | **70.76** | 69.65 | 59.17 | 69.00 | 67.60 | 54.02 | **66.94** | 65.42 |
| | | | 1 | 0.5 | 53.12 | 73.51 | 70.61 | 48.51 | 71.55 | 67.67 | 45.47 | 69.55 | 65.74 |
| | | | 1 | 0 | 60.84 | 72.47 | 71.18 | 58.19 | **68.94** | 67.72 | 55.27 | 67.76 | 65.71 |
| | $\epsilon_{\text{test}} = \frac{16}{255}$ $\epsilon_{\text{train}} = \frac{17.6}{255}$ [3] | IBP | 0 | 0 | 70.02 | 78.86 | 77.67 | 67.55 | 78.65 | 76.92 | 68.97 | 78.12 | 76.66 |
| | | | 1 | 0.5 | 63.43 | 81.58 | 78.81 | 60.07 | 81.01 | 77.32 | 59.46 | 80.85 | 76.97 |
| | | | 1 | 0 | 67.73 | 78.71 | 77.52 | 70.28 | 79.26 | 77.43 | 68.88 | 78.91 | 76.95 |
| | | CROWN-IBP | 0 | 0 | 67.42 | **78.41** | 76.86 | 68.06 | 77.92 | 76.89 | 67.17 | 77.27 | 75.76 |
| | | | 1 | 0.5 | 61.47 | 79.62 | 77.13 | 59.56 | 79.30 | 76.43 | 56.73 | 78.20 | 74.87 |
| | | | 1 | 0 | 68.75 | 78.71 | 77.91 | 67.94 | **78.46** | 77.21 | 66.06 | **76.80** | 75.23 |

[1] Verified errors reported in Table 4 of Gowal et al. (2018) are evaluated using mixed integer programming (MIP). For a fair comparison, we use the IBP verified errors reported in Table 3 of Gowal et al. (2018).

[2] According to direct communication with the authors of Gowal et al. (2018), achieving 68.44% IBP verified error requires to adding an extra PGD adversarial training loss. Without adding PGD, the achievable verified error is 72.91% (LP/MIP verified) or 73.52% (IBP verified).

[3] Although not explicitly mentioned, the best CIFAR-10 models in (Gowal et al., 2018) also use $\epsilon_{\text{train}} = 1.1\epsilon_{\text{test}}$.

[4] We use $\beta_{\text{start}} = \beta_{\text{end}} = 1$ for this setting, the same as in Table 2, and thus CROWN-IBP bound is used to evaluate the verified error.

Table D: Verified and standard (clean) test errors for a large number of models trained on MNIST and CIFAR-10 datasets using IBP and CROWN-IBP. The purpose of this experiment is to compare model performance statistics (min, median and max) *on a wide range of models*, rather than a few hand selected models. For each setting we report 3 representative models: the models with smallest, median, and largest verified error. We also report the standard error of these three selected models. Note that in this table we set $\epsilon_{\text{train}} = \epsilon_{\text{test}}$ and observe overfitting on small $\epsilon$ for MNIST. See Section I for detailed discussions.

| Dataset | $\epsilon$ ($\ell_\infty$ norm) | Model Family | Training Method | $\kappa_{\text{start}}$ | $\kappa_{\text{end}}$ | Verified Test Error (%) best | median | worst | Standard Test Error(%) best | median | worst |
|---|---|---|---|---|---|---|---|---|---|---|---|
| MNIST | $\epsilon_{\text{train}} = 0.1$ $\epsilon_{\text{test}} = 0.1$ | 10 small models | IBP | 0 | 0 | 4.79 | 5.74 | 7.32 | 1.48 | 1.59 | 2.50 |
| | | | | 1 | 0 | 4.87 | 5.72 | 7.24 | 1.51 | 1.34 | 2.46 |
| | | | | 1 | 0.5 | 5.24 | 5.95 | 7.36 | 1.41 | 1.88 | 1.87 |
| | | | CROWN-IBP | 0 | 0 | 4.21 | **5.18** | **6.80** | 1.41 | 1.83 | 2.58 |
| | | | | 1 | 0 | **4.14** | 5.24 | 6.82 | 1.39 | 2.06 | 2.46 |
| | | | | 1 | 0.5 | 4.62 | 5.94 | 6.88 | 1.26 | 1.88 | 1.97 |
| | | 8 medium models | IBP | 0 | 0 | 5.9 | 6.25 | 7.82 | 1.14 | 1.12 | 1.23 |
| | | | | 1 | 0 | 5.77 | 6.30 | 7.50 | 1.21 | 1.13 | 1.34 |
| | | | | 1 | 0.5 | 6.05 | 6.40 | 7.70 | 1.19 | 1.33 | 1.24 |
| | | | CROWN-IBP | 0 | 0 | **5.22** | **5.63** | 6.34 | 1.19 | 1.05 | 1.03 |
| | | | | 1 | 0 | 5.43 | 5.90 | **6.02** | 1.30 | 1.03 | 1.09 |
| | | | | 1 | 0.5 | 5.44 | 5.89 | 6.09 | 1.11 | 1.16 | 1.01 |
| | $\epsilon_{\text{train}} = 0.2$ $\epsilon_{\text{test}} = 0.2$ | 10 small models | IBP | 0 | 0 | 6.90 | 8.24 | 12.67 | 1.93 | 2.76 | 4.14 |
| | | | | 1 | 0 | 6.84 | 8.16 | 12.92 | 2.01 | 2.56 | 3.93 |
| | | | | 1 | 0.5 | 7.31 | 8.71 | 13.54 | 1.62 | 2.36 | 3.22 |
| | | | CROWN-IBP | 0 | 0 | **6.11** | 7.29 | **11.97** | 1.93 | 2.3 | 3.86 |
| | | | | 1 | 0 | 6.27 | 7.66 | 12.11 | 2.01 | 2.92 | 4.06 |
| | | | | 1 | 0.5 | 6.53 | 8.14 | 12.56 | 1.61 | 1.61 | 3.27 |
| | | 8 medium models | IBP | 0 | 0 | 7.56 | 8.60 | 9.80 | 1.96 | 2.19 | 1.39 |
| | | | | 1 | 0 | 8.26 | 8.72 | 9.84 | 1.45 | 1.73 | 1.31 |
| | | | | 1 | 0.5 | 8.42 | 8.90 | 10.09 | 1.76 | 1.42 | 1.53 |
| | | | CROWN-IBP | 0 | 0 | **6.06** | **6.42** | **7.64** | 1.09 | 1.33 | 1.36 |
| | | | | 1 | 0 | 6.39 | 7.09 | 7.84 | 1.11 | 1.04 | 1.25 |
| | | | | 1 | 0.5 | 6.63 | 7.51 | 7.96 | 1.08 | 1.25 | 1.19 |
| | $\epsilon_{\text{train}} = 0.3$ $\epsilon_{\text{test}} = 0.3$ | 10 small models | IBP | 0 | 0 | 10.54 | 12.02 | 20.47 | 2.78 | 3.31 | 6.07 |
| | | | | 1 | 0 | 9.96 | 12.09 | 21.0 | 2.7 | 3.48 | 6.68 |
| | | | | 1 | 0.5 | 10.37 | 12.78 | 21.99 | 2.11 | 3.44 | 5.19 |
| | | | CROWN-IBP | 0 | 0 | **8.87** | **11.29** | **16.83** | 2.43 | 3.62 | 7.26 |
| | | | | 1 | 0 | 9.69 | 11.33 | 15.23 | 2.78 | 3.41 | 5.90 |
| | | | | 1 | 0.5 | 9.90 | 11.98 | 19.56 | 2.20 | 2.72 | 4.83 |
| | | 8 medium models | IBP | 0 | 0 | 10.43 | 10.83 | 11.99 | 2.01 | 2.38 | 3.29 |
| | | | | 1 | 0 | 10.74 | 11.73 | 12.16 | 2.17 | 2.46 | 1.60 |
| | | | | 1 | 0.5 | 11.23 | 11.71 | 12.4 | 1.72 | 2.09 | 1.63 |
| | | | CROWN-IBP | 0 | 0 | **7.46** | **8.47** | **8.57** | 1.48 | 1.52 | 1.99 |
| | | | | 1 | 0 | 7.96 | 8.53 | 8.99 | 1.45 | 1.56 | 1.85 |
| | | | | 1 | 0.5 | 8.19 | 9.20 | 9.51 | 1.27 | 1.46 | 1.62 |
| | $\epsilon_{\text{train}} = 0.4$ $\epsilon_{\text{test}} = 0.4$ | 10 small models | IBP | 0 | 0 | 16.72 | 18.89 | 37.42 | 4.2 | 5.4 | 9.63 |
| | | | | 1 | 0 | 16.10 | 18.75 | 35.3 | 3.8 | 4.93 | 11.32 |
| | | | | 1 | 0.5 | 16.54 | 19.14 | 35.42 | 3.40 | 3.65 | 7.54 |
| | | | CROWN-IBP | 0 | 0 | **15.38** | **18.57** | **24.56** | 3.61 | 4.83 | 8.46 |
| | | | | 1 | 0 | 16.22 | 18.20 | 24.80 | 4.23 | 5.15 | 8.54 |
| | | | | 1 | 0.5 | 15.97 | 19.18 | 24.76 | 3.48 | 3.97 | 6.64 |
| | | 8 medium models | IBP | 0 | 0 | 15.17 | 16.54 | 18.98 | 2.83 | 3.79 | 4.91 |
| | | | | 1 | 0 | 15.63 | 16.06 | 17.11 | 2.93 | 3.4 | 3.75 |
| | | | | 1 | 0.5 | 15.74 | 16.42 | 17.98 | 2.35 | 2.31 | 3.15 |
| | | | CROWN-IBP | 0 | 0 | 12.96 | **13.43** | 14.25 | 2.76 | 2.85 | 3.36 |
| | | | | 1 | 0 | **12.90** | 13.47 | **14.06** | 2.42 | 2.86 | 3.11 |
| | | | | 1 | 0.5 | 13.02 | 13.69 | 14.52 | 1.89 | 2.40 | 2.35 |
| CIFAR-10 | $\epsilon_{\text{train}} = 2/255$[2] $\epsilon_{\text{test}} = 2/255$ | 9 small models | IBP | 0 | 0 | 54.69 | 57.84 | 60.58 | 40.59 | 45.51 | 51.38 |
| | | | | 1 | 0 | 54.56 | 58.42 | 60.69 | 40.32 | 47.42 | 50.73 |
| | | | | 1 | 0.5 | 56.89 | 60.66 | 63.58 | 34.28 | 39.28 | 48.03 |
| | | | CROWN-IBP | 0 | 0 | **48.87** | **54.68** | **58.82** | 32.20 | 40.08 | 46.98 |
| | | | | 1 | 0 | 49.45 | 55.09 | 59.00 | 32.22 | 40.45 | 47.05 |
| | | | | 1 | 0.5 | 52.14 | 57.49 | 60.12 | 28.03 | 35.76 | 43.40 |
| | | 8 medium models | IBP | 0 | 0 | 55.47 | 56.41 | 58.54 | 41.59 | 44.33 | 46.54 |
| | | | | 1 | 0 | 55.51 | 56.74 | 57.85 | 42.41 | 43.71 | 44.74 |
| | | | | 1 | 0.5 | 57.05 | 59.70 | 60.25 | 34.77 | 35.80 | 38.95 |
| | | | CROWN-IBP | 0 | 0 | 49.57 | **50.83** | **52.59** | 32.64 | 34.20 | 37.06 |
| | | | | 1 | 0 | **49.28** | 51.59 | 53.45 | 32.31 | 34.23 | 38.11 |
| | | | | 1 | 0.5 | 52.05 | 53.56 | 55.23 | 27.80 | 29.49 | 32.42 |
| | $\epsilon_{\text{train}} = 8/255$ $\epsilon_{\text{test}} = 8/255$ | 9 small models | IBP | 0 | 0 | 72.07 | 73.34 | 73.88 | 61.11 | 61.01 | 64.0 |
| | | | | 1 | 0 | 72.42 | 72.57 | 73.49 | 62.26 | 60.98 | 63.5 |
| | | | | 1 | 0.5 | 73.88 | 75.16 | 76.93 | 55.66 | 52.53 | 53.79 |
| | | | CROWN-IBP | 0 | 0 | 71.28 | **72.15** | 73.66 | 59.40 | 60.80 | 63.10 |
| | | | | 1 | 0 | **70.77** | 72.24 | **73.10** | 58.65 | 60.49 | 61.86 |
| | | | | 1 | 0.5 | 72.59 | 74.71 | 76.11 | 49.86 | 52.95 | 55.58 |
| | | 8 medium models | IBP | 0 | 0 | 72.75 | 73.23 | 73.82 | 59.23 | 65.96 | 66.35 |
| | | | | 1 | 0 | 72.18 | 72.83 | 74.38 | 62.54 | 59.6 | 61.99 |
| | | | | 1 | 0.5 | 74.84 | 75.59 | 97.93 | 51.71 | 54.41 | 54.12 |
| | | | CROWN-IBP | 0 | 0 | 70.79 | **71.61** | **72.29** | 57.90 | 60.10 | 59.70 |
| | | | | 1 | 0 | **70.51** | 71.96 | 72.82 | 57.87 | 59.98 | 59.16 |
| | | | | 1 | 0.5 | 73.48 | 74.69 | 76.66 | 49.40 | 53.56 | 52.05 |

[1] Verified errors reported in Table 4 of Gowal et al. (2018) are evaluated using mixed integer programming (MIP) and linear programming (LP), which are strictly smaller than IBP verified errors but computationally expensive. For a fair comparison, we use the IBP verified errors reported in their Table 3.

[2] We use $\beta_{\text{start}} = \beta_{\text{end}} = 1$ for this setting, the same as in Table 2, and thus CROWN-IBP bound is used to evaluate the verified error.

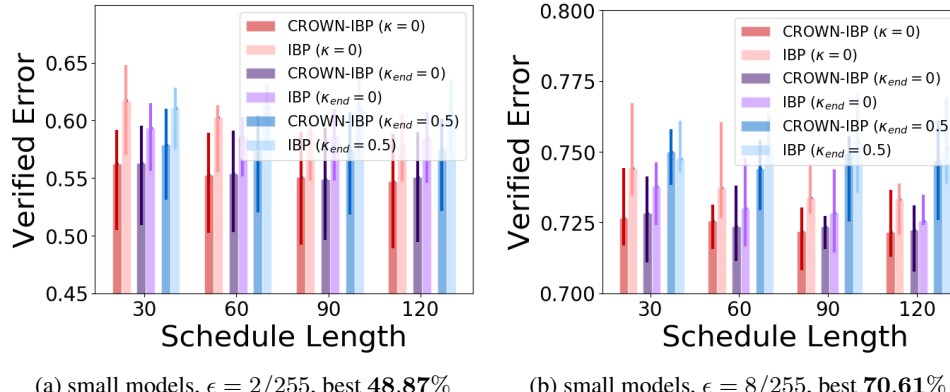

(a) small models, $\epsilon = 2/255$, best **48.87%**          (b) small models, $\epsilon = 8/255$, best **70.61%**

Figure C: Verified error vs. schedule length (30, 60, 90, 120) on 9 small models on CIFAR-10. The solid boxes show median values of verified errors. $\kappa_{\text{start}} = 1.0$ except for the $\kappa = 0$ setting. The upper and lower bound of an error bar are worst and best verified error, respectively.

## G  REPRODUCIBILITY

To further test the training stability of CROWN-IBP, we run each MNIST experiment (using selected models in Table B) 5 times to get the mean and standard deviation of the verified and standard errors on test set. Results are presented in Table E. Standard deviations of verified errors are very small, giving us further evidence of good stability and reproduciblity.

| $\epsilon$ | error | model A | model B | model C | model D | model E | model F | model G | model H | model I | model J |
|---|---|---|---|---|---|---|---|---|---|---|---|
| 0.1 | std. err. (%) | $2.57 \pm .04$ | $1.45 \pm .05$ | $3.02 \pm .04$ | $1.77 \pm .04$ | $2.13 \pm .08$ | $1.35 \pm .05$ | $2.03 \pm .08$ | $1.32 \pm .08$ | $1.77 \pm .04$ | $1.45 \pm .05$ |
| | verified err. (%) | $6.85 \pm .04$ | $4.88 \pm .04$ | $6.67 \pm .1$ | $5.10 \pm .1$ | $4.82 \pm .2$ | $4.18 \pm .008$ | $5.23 \pm .2$ | $4.59 \pm .08$ | $5.92 \pm .09$ | $5.40 \pm .09$ |
| 0.2 | std. err. (%) | $3.87 \pm .04$ | $2.43 \pm .04$ | $4.40 \pm .2$ | $2.32 \pm .04$ | $3.45 \pm .3$ | $1.90 \pm 0$ | $2.67 \pm .1$ | $2.00 \pm .07$ | $2.22 \pm .04$ | $1.65 \pm .05$ |
| | verified err. (%) | $12.0 \pm .03$ | $6.99 \pm .04$ | $10.3 \pm .2$ | $7.37 \pm .06$ | $9.01 \pm .9$ | $6.05 \pm .03$ | $7.50 \pm .1$ | $6.45 \pm .06$ | $7.50 \pm .3$ | $6.31 \pm .08$ |
| 0.3 | std. err. (%) | $5.97 \pm .08$ | $3.20 \pm 0$ | $6.78 \pm .1$ | $3.70 \pm .1$ | $3.85 \pm .2$ | $3.10 \pm .1$ | $4.20 \pm .3$ | $2.85 \pm .05$ | $3.67 \pm .08$ | $2.35 \pm .09$ |
| | verified err. (%) | $15.4 \pm .08$ | $10.6 \pm .06$ | $16.1 \pm .3$ | $11.3 \pm .1$ | $11.7 \pm .2$ | $9.96 \pm .09$ | $12.2 \pm .6$ | $9.90 \pm .2$ | $11.2 \pm .09$ | $9.21 \pm .3$ |
| 0.4 | std. err. (%) | $8.43 \pm .04$ | $4.93 \pm .1$ | $8.53 \pm .2$ | $5.83 \pm .2$ | $5.48 \pm .2$ | $4.65 \pm .09$ | $6.80 \pm .2$ | $4.28 \pm .1$ | $5.60 \pm .1$ | $3.60 \pm .07$ |
| | verified err. (%) | $24.6 \pm .1$ | $18.5 \pm .2$ | $24.6 \pm .7$ | $19.2 \pm .2$ | $18.8 \pm .2$ | $17.3 \pm .04$ | $20.4 \pm .3$ | $16.3 \pm .2$ | $18.5 \pm .07$ | $15.2 \pm .3$ |

Table E: Means and standard deviations of verified and standard errors of 10 MNIST models trained using CROWN-IBP. The architectures of these models are presented in Table B. We run each model 5 times to compute its mean and standard deviation.

## H  TRAINING STABILITY EXPERIMENTS ON OTHER $\epsilon$

Similar to our experiments in Section 4, we compare the verified errors obtained by CROWN-IBP and IBP under different $\epsilon$ schedule lengths (10, 15, 30, 60) on MNIST and (30,60,90,120) on CIFAR-10. We present the best, worst and median verified errors over all 18 models for MNIST in Figure D, E at $\epsilon \in \{0.1, 0.2, 0.3\}$ and 9 small models for CIFAR-10 in Figure C. The upper and lower ends of an error bar are the worst and best verified error, respectively, and the solid boxes represent median values. CROWN-IBP can improve training stability, and consistently outperform IBP under different schedule length and $\kappa$ settings.

## I  OVERFITTING ISSUE WITH SMALL $\epsilon$

We found that on MNIST for a small $\epsilon$, the verified error obtained by IBP based methods are not as good as linear relaxation based methods (Wong et al., 2018; Mirman et al., 2018). Gowal et al. (2018) thus propose to train models using a larger $\epsilon$ and evaluate them under a smaller $\epsilon$, for example $\epsilon_{\text{train}} = 0.4$ and $\epsilon_{\text{eval}} = 0.3$. Instead, we investigated this issue further and found that many CROWN-IBP trained models achieve very small verified errors (close to 0 and sometimes exactly 0) on training set (see Table F). This indicates possible overfitting during training. As we discussed in Section 3, linear relaxation based methods implicitly regularize the weight matrices so the network does not overfit when $\epsilon$ is small. Inspired by this finding, we want to see if adding an explicit $\ell_1$

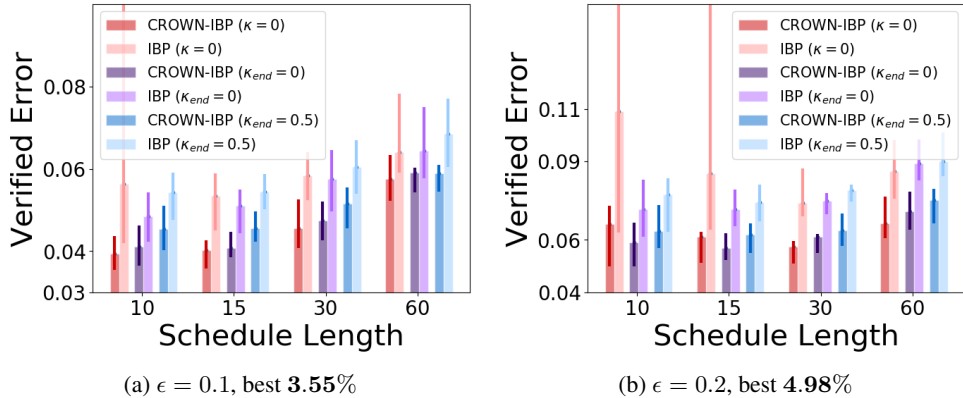

(a) $\epsilon = 0.1$, best **3.55%**          (b) $\epsilon = 0.2$, best **4.98%**

Figure D: Verified error vs. $\epsilon$ schedule length (10, 15, 30, 60) on 8 medium MNIST models. The upper and lower ends of a vertical bar represent the worst and best verified error, respectively. The solid boxes represent the median values of the verified error. For a small $\epsilon$, using a shorter schedule length improves verified error due to early stopping, which prevents overfitting. All best verified errors are achieved by CROWN-IBP regardless of schedule length.

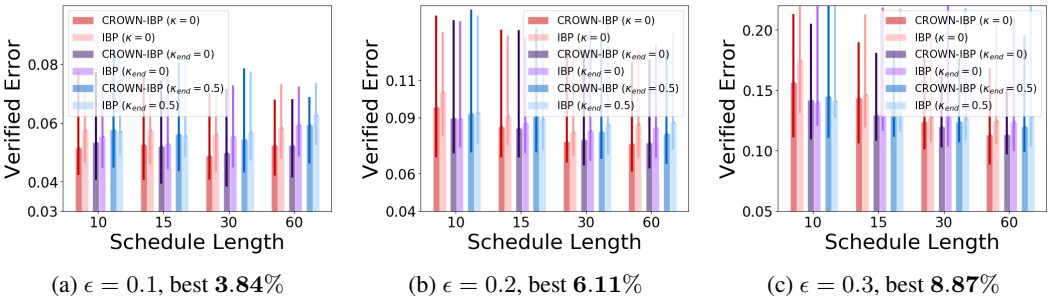

(a) $\epsilon = 0.1$, best **3.84%**          (b) $\epsilon = 0.2$, best **6.11%**          (c) $\epsilon = 0.3$, best **8.87%**

Figure E: Verified error vs. $\epsilon$ schedule length (10, 15, 30, 60) on 10 small MNIST models. The upper and lower ends of a vertical bar represent the worst and best verified error, respectively. All best verified errors are achieved by CROWN-IBP regardless of schedule length.

regularization term in CROWN-IBP training helps when $\epsilon_{\text{train}} = 0.1$ or $0.2$. The verified and standard errors on the training and test sets with and without regularization can be found in Table F. We can see that with a small $\ell_1$ regularization added ($\lambda = 5 \times 10^{-5}$) we can reduce verified errors on test set significantly. This makes CROWN-IBP results comparable to the numbers reported in convex adversarial polytope (Wong et al., 2018); at $\epsilon = 0.1$, the best model using convex adversarial polytope training can achieve $3.67\%$ verified error, while CROWN-IBP achieves $3.60\%$ best certified error on the models presented in Table F. The overfitting is likely caused by IBP's strong learning power without over-regularization, which also explains why IBP based methods significantly outperform linear relaxation based methods at larger $\epsilon$ values. Using early stopping can also improve verified error on test set; see Figure D.

## J    TRAINING TIME

In Table G we present the training time of CROWN-IBP, IBP and convex adversarial polytope (Wong et al., 2018) on several representative models. All experiments are measured on a single RTX 2080 Ti GPU with 11 GB RAM except for 2 DM-Large models where we use 4 RTX 2080 Ti GPUs to speed up training. We can observe that CROWN-IBP is practically 1.5 to 3.5 times slower than IBP. Theoretically, CROWN-IBP is up to $n_L = 10$ times slower[4] than IBP; however usually the total training time is less than 10 times since the CROWN-IBP bound is only computed during the ramp-up phase, and CROWN-IBP has higher GPU computation intensity and thus better GPU utilization than IBP. convex adversarial polytope (Wong et al., 2018), as a representative linear relaxation based

---

[4]More precisely, $n_L - 1 = 9$ times slower as we can omit the all-zero row in specification matrix Eq. (3).

| $\epsilon$ | Model Name (see Appendix D) | $\lambda$: $\ell_1$ regularization | Training | | Test | |
|---|---|---|---|---|---|---|
| | | | standard error | verified error | standard error | verified error |
| 0.1 | P | 0 | 0.01% | 0.01% | 1.05% | 5.63% |
| | P | $5 \times 10^{-5}$ | 0.32% | 0.98% | 1.30% | **3.60%** |
| | O | 0 | 0.02% | 0.05% | 0.82% | 6.02% |
| | O | $5 \times 10^{-5}$ | 0.38% | 1.34% | 1.43% | 4.02% |
| 0.2 | P | 0 | 0.35% | 1.40% | 1.09% | 6.06% |
| | P | $5 \times 10^{-5}$ | 1.02% | 3.73% | 1.48% | **5.48%** |
| | O | 0 | 0.31% | 1.54% | 1.22% | 6.64% |
| | O | $5 \times 10^{-5}$ | 1.09% | 4.08% | 1.69% | 5.72% |

Table F: $\ell_1$ regularized and unregularized models' standard and verified errors on training and test set. At a small $\epsilon$, CROWN-IBP may overfit and adding regularization helps robust generalization; on the other hand, convex relaxation based methods (Wong et al., 2018) provides implicitly regularization which helps generalization under small $\epsilon$ but deteriorate model performance at larger $\epsilon$.

method, can be over hundreds times slower than IBP especially on deeper networks. Note that we use 50 random Cauchy projections for (Wong et al., 2018). Using random projections alone is not sufficient to scale purely linear relaxation based methods to larger datasets, thus we advocate a combination of IBP bounds with linear relaxation based methods as in CROWN-IBP, which offers good scalability and stability. We also note that the random projection based acceleration can also be applied to the backward bound propagation (CROWN-style bound) in CROWN-IBP to further speed up CROWN-IBP.

Table G: IBP and CROWN-IBP's training time on different models in seconds. For IBP and CROWN-IBP, we use a batchsize of 256 for MNIST and 128 for CIFAR-10. For convex adversarial polytope, we use 50 random Cauchy projections, and reduce batch size if necessary to fit into GPU memory.

| Data | MNIST | | | | | | CIFAR-10 | | | | | |
|---|---|---|---|---|---|---|---|---|---|---|---|---|
| Model Name | A | C | G | L | O | DM-large ($\epsilon_{\text{train}} = 0.4$) | B | D | H | S | M | DM-large |
| IBP (s) | 245 | 264 | 290 | 364 | 1032 | 3769[1] | 734 | 908 | 1048 | 691 | 1407 | 40496[1] |
| CROWN-IBP (s) | 371 | 564 | 590 | 954 | 3649 | 5584[1] | 1148 | 1853 | 1859 | 1491 | 4137 | 91288[1] |
| CAP (Wong et al., 2018)[2](s) | 1708 | 9263 | 12649 | 35518 | 160794 | — | 2372 | 12688 | 18691 | 6961 | 51145 | — |

[1] We use 4 GPUs to train this model.

[2] Convex adversarial polytopes (CAP) are computed with 50 random projections. Without random projections it will not scale to most models except for the smallest ones.

# K    REPRODUCING CIFAR-10 RESULTS ON MULTI-GPUS

The use of 32 TPUs for our CIFAR-10 experiments is not necessary. We use TPUs mainly for obtaining a completely fair comparison to IBP (Gowal et al., 2018), as their implementation was TPU-based. Since TPUs are not widely available, we additionally implemented CROWN-IBP using multi-GPUs. We train the best models in Table 2 on 4 RTX 2080Ti GPUs. As shown in Table H, we can achieve comparable verified errors using GPUs, and the differences between GPU and TPU training are around $\pm 0.5\%$. Training time is reported in Table G.

# L    EXACT FORMS OF THE CROWN-IBP BACKWARD BOUND

CROWN (Zhang et al., 2018) is a general framework that replaces non-linear functions in a neural network with linear upper and lower hyperplanes with respect to pre-activation variables, such that the entire neural network function can be bounded by a linear upper hyperplane and linear lower hyperplane for all $\boldsymbol{x} \in S$ ($S$ is typically a norm bounded ball, or a box region):

$$\underline{\mathbf{A}}x + \underline{\mathbf{b}} \leq f(\boldsymbol{x}) \leq \overline{\mathbf{A}}x + \overline{\mathbf{b}}$$

CROWN achieves such linear bounds by replacing non-linear functions with linear bounds, and utilizing the fact that the linear combinations of linear bounds are still linear, thus these linear bounds

Table H: Comparison of verified and standard errors for CROWN-IBP models trained on TPUs and GPUs (CIFAR-10, DM-Large model).

| Dataset | $\epsilon$ ($\ell_\infty$ norm) | Training Device | $\kappa$ schedules | | Model errors (%) | |
| --- | --- | --- | --- | --- | --- | --- |
| | | | $\kappa_{\text{start}}$ | $\kappa_{\text{end}}$ | Standard | **Verified** |
| CIFAR-10 | $\epsilon_{\text{test}} = \frac{2}{255}$ [1] $\epsilon_{\text{train}} = \frac{2.2}{255}$ | GPU | 0 | 0 | 29.18 | 45.50 |
| | | TPU | 0 | 0 | 28.48 | 46.03 |
| | $\epsilon_{\text{test}} = \frac{8}{255}$ $\epsilon_{\text{train}} = \frac{8.8}{255}$ | GPU | 0 | 0 | 54.60 | 67.11 |
| | | TPU | 0 | 0 | 54.02 | 66.94 |

[1] We use $\beta_{\text{start}} = \beta_{\text{end}} = 1$ for this setting, the same as in Table 2, and thus CROWN-IBP bound is used to evaluate the verified error.

can propagate through layers. Suppose we have a non-linear vector function $\sigma$, applying to an input (pre-activation) vector $z$, CROWN requires the following bounds in a general form:

$$\underline{\mathbf{A}}_\sigma z + \underline{\mathbf{b}}_\sigma \leq \sigma(z) \leq \overline{\mathbf{A}}_\sigma z + \overline{\mathbf{b}}_\sigma$$

In general the specific bounds $\underline{\mathbf{A}}_\sigma, \underline{\mathbf{b}}_\sigma, \overline{\mathbf{A}}_\sigma, \overline{\mathbf{b}}_\sigma$ for different $\sigma$ needs to be given in a case-by-case basis, depending on the characteristics of $\sigma$ and the preactivation range $\underline{z} \leq z \leq \overline{z}$. In neural network common $\sigma$ can be ReLU, tanh, sigmoid, maxpool, etc. Convex adversarial polytope (Wong et al., 2018) is also a linear relaxation based techniques that is closely related to CROWN, but only for ReLU layers. For ReLU such bounds are simple, where $\underline{\mathbf{A}}_\sigma, \overline{\mathbf{A}}_\sigma$ are diagonal matrices, $\underline{\mathbf{b}}_\sigma = \mathbf{0}$:

$$\underline{\mathbf{D}}z \leq \sigma(z) \leq \overline{\mathbf{D}}z + \overline{c} \tag{14}$$

where $\underline{\mathbf{D}}$ and $\overline{\mathbf{D}}$ are two diagonal matrices:

$$\underline{\mathbf{D}}_{k,k} = \begin{cases} 1, & \text{if } \underline{z}_k > 0, \text{ i.e., this neuron is always active} \\ 0, & \text{if } \overline{z}_k < 0, \text{ i.e., this neuron is always inactive} \\ \alpha, & \text{otherwise, any } 0 \leq \alpha \leq 1 \end{cases} \tag{15}$$

$$\overline{\mathbf{D}}_{k,k} = \begin{cases} 1, & \text{if } \underline{z}_k > 0, \text{ i.e., this neuron is always active} \\ 0, & \text{if } \overline{z}_k < 0, \text{ i.e., this neuron is always inactive} \\ \frac{\overline{z}_k}{\overline{z}_k - \underline{z}_k}, & \text{otherwise} \end{cases} \tag{16}$$

$$\overline{c}_k = \begin{cases} 0, & \text{if } \underline{z}_k > 0, \text{ i.e., this neuron is always active} \\ 0, & \text{if } \overline{z}_k < 0, \text{ i.e., this neuron is always inactive} \\ \frac{\overline{z}_k \underline{z}_k}{\overline{z}_k - \underline{z}_k}, & \text{otherwise} \end{cases} \tag{17}$$

Note that CROWN-style bounds require to know all pre-activation bounds $\underline{z}^{(l)}$ and $\overline{z}^{(l)}$. We assume these bounds are valid for $x \in S$. In CROWN-IBP, these bounds are obtained by interval bound propagation (IBP). With pre-activation bounds $\underline{z}^{(l)}$ and $\overline{z}^{(l)}$ given (for $x \in S$), we rewrite the CROWN lower bound for the special case of ReLU neurons:

**Theorem L.1** (CROWN Lower Bound). *For a $L$-layer neural network function $f(x) : \mathbb{R}^{n_0} \to \mathbb{R}^{n_L}$, $\forall j \in [n_L]$, $\forall x \in S$, we have $\underline{f}_j(x) \leq f_j(x)$, where*

$$\underline{f}_j(x) = \mathbf{A}_{j,:}^{(0)} x + \sum_{l=1}^{L} \mathbf{A}_{j,:}^{(l)}(b^{(l)} + \underline{b}^{j,(l)}), \tag{18}$$

$$\mathbf{A}_{j,:}^{(l)} = \begin{cases} \mathbf{e}_j^\top & \text{if } l = L; \\ \mathbf{A}_{j,:}^{(l+1)} \mathbf{W}^{(l+1)} \mathbf{D}^{j,(l)} & \text{if } l \in \{0, \cdots, L-1\}. \end{cases}$$

*and $\forall i \in [n_k]$, we define diagonal matrices $\mathbf{D}^{j,(l)}$, bias vector $\underline{\mathbf{b}}^{(l)}$:*

$$\mathbf{D}^{j,(0)} = \boldsymbol{I}, \quad \underline{\mathbf{b}}^{j,(L)} = \mathbf{0}$$

$$\mathbf{D}_{k,k}^{j,(l)} = \begin{cases} 1 & \text{if } \mathbf{A}_{j,:}^{(l+1)}\mathbf{W}_{:,i}^{(l+1)} \geq 0, \overline{z}_k^{(l)} > |\underline{z}_k^{(l)}|, \ l \in \{1, \cdots, L-1\}; \\ 0 & \text{if } \mathbf{A}_{j,:}^{(l+1)}\mathbf{W}_{:,i}^{(l+1)} \geq 0, \overline{z}_k^{(l)} < |\underline{z}_k^{(l)}|, \ l \in \{1, \cdots, L-1\}; \\ \frac{\overline{z}_k^{(l)}}{\overline{z}_k^{(l)} - \underline{z}_k^{(l)}} & \text{if } \mathbf{A}_{j,:}^{(k+1)}\mathbf{W}_{:,i}^{(k+1)} < 0, \ l \in \{1, \cdots, L-1\}. \end{cases}$$

$$\underline{\mathbf{b}}_k^{j,(l)} = \begin{cases} 0 & \text{if } \mathbf{A}_{j,:}^{(l+1)}\mathbf{W}_{:,i}^{(l+1)} \geq 0; \ l \in \{1, \cdots, L-1\} \\ \frac{\overline{z}_k^{(l)}\underline{z}_k^{(l)}}{\overline{z}_k^{(l)} - \underline{z}_k^{(l)}} & \text{if } \mathbf{A}_{j,:}^{(l+1)}\mathbf{W}_{:,i}^{(l+1)} < 0 \ l \in \{1, \cdots, L-1\}. \end{cases}$$

$\mathbf{e}_j \in \mathbb{R}^{n_L}$ *is a standard unit vector with $j$-th coordinate set to 1.*

Note that unlike the ordinary CROWN (Zhang et al., 2018), in CROWN-IBP we only need the lower bound to compute $\underline{m}$ and do not need to compute the $\mathbf{A}$ matrices for the upper bound. This save half of the computation cost in ordinary CROWN. Also, $\mathbf{W}$ represents any affine layers in a neural network, including convolutional layers in CNNs. In Section 3.2, we discussed how to use transposed convolution operators to efficiently implement CROWN-IBP on GPUs.

Although in this paper we focus on the common case of ReLU activation function, other general activation functions (sigmoid, max-pooling, etc) can be used in the network as CROWN is a general framework to deal with non-linearity. For a more general derivation we refer the readers to (Zhang et al., 2018) and (Salman et al., 2019b).

