# OpenReview forum: "Towards Stable and Efficient Training of Verifiably Robust Neural Networks"
_ICLR.cc/2020/Conference — Accept (Poster)_

### Official Review · AnonReviewer2 · 2019-10-25
**Official Blind Review #2**

**Rating:** 6

**Review:**

This paper proposes a new method for training certifiably robust models that achieves better results than the previous SOTA results by IBP, with a moderate increase in training time. It uses a CROWN-based bound in the warm up phase of IBP, which serves as a better initialization for the later phase of IBP and lead to improvements in both robust and standard accuracy. The CROWN-based bound uses IBP to compute bounds for intermediate pre-activations and applies CROWN only to computing the bounds of the margins, which has a complexity between IBP and CROWN. The experimental results are verify detailed to demonstrate the improvement.

The improvement is significant enough to me and I tend to accept the paper. The results on CIFAR10 with epsilon=8/255 is so far the state-of-the-art. However, it is far from being scalable enough to large networks and datasets, which has already been achieved by randomized smoothing based approaches. On CIFAR10, it takes 32 TPU cores to train a 4-conv-layer network. Still, such an approach has the advantage of making robust inferences much more efficiently than randomized smoothing, and thus still worth further explorations.

**Experience Assessment:**

I have published one or two papers in this area.

**Review Assessment: Checking Correctness Of Derivations And Theory:**

I assessed the sensibility of the derivations and theory.

**Review Assessment: Checking Correctness Of Experiments:**

I carefully checked the experiments.

**Review Assessment: Thoroughness In Paper Reading:**

I read the paper at least twice and used my best judgement in assessing the paper.

---

> ### Author Response · Authors · 2019-11-09
> **Thank you for the discussions on randomized smoothing; TPUs are not necessary for CROWN-IBP (Multi-GPU version will be released)**
>
> Dear AnonReviewer2,
>
> We thank you for recognizing the contributions of our paper and raising the discussions on randomized smoothing and concerns on expensive computations on TPUs.
>
> (Answer 1) Compared to the randomized smoothing based method, our bound propagation-based approach has several theoretical and practical benefits:
>
> 1. Recent works [1][2] show that randomized smoothing may not scale well for the important case of L infinity robustness. They provided some preliminary theoretical evidence that even the *optimal* robustness certificate for L infinity smoothing has a dependency on dimension d, thus for high dimensional input (e.g., CIFAR with d=3072), randomized smoothing based method cannot give a good quality bound. On the other hand, for L2 norm, the randomized smoothing certificate is dimension-free. This is a fundamental limitation of randomized smoothing. For the L infinity setting like CIFAR epsilon=8/255, bound propagation-based method like CROWN-IBP still gives the best results.
>
> 2. As also mentioned by the reviewer, randomized smoothing typically needs a large number of samples, e.g., in Cohen et al., 100,000 random samples for a *single* image. In contrast, our verification can be computed using IBP very fast, which is only 2x forward propagation time. Randomized smoothing costs 50,000x more during inference, and our training procedure is 500x (pessimistically) slower during training time. So it is really a trade-off here; each method has its own strength.
>
> Being scalable to large networks on all important norms with less training/inference cost is still an open challenge. It is not solved by randomized smoothing, nor CROWN-IBP. For the next step, our future work will investigate how to combine the strengths from bound propagation-based certified defense (good for L infinity norm, sample free) and randomized smoothing based approach (good for L2 norm, need a lot of samples). Thus, our contribution as the SOTA bound propagation-based certified defense is important, as it can become an ingredient of the next generation certified defense.
>
> (Answer 2) Regarding computation cost, the use of 32 TPUs is not necessary. We use TPUs mainly for obtaining a completely fair comparison to IBP (Gowal et al, 2018), as their implementation was TPU-based. We additionally implemented CROWN-IBP using multi-GPUs. Training the same largest CIFAR network takes 1 day on 4x 1080 Ti GPUs (using the same hyperparameters), and we can achieve similar accuracy. We think this computational cost is quite reasonable, compared to other SOTA uncertified defense like adversarial training, which is also quite slow (10-20x extra cost for each epoch, and needs much more epochs to converge than natural training).
>
> We updated new multi-GPU training results in Table H, and we will open source our multi-GPU training code to make our algorithm available to a broader audience.
>
> We hope our response addresses your concerns on TPU training and randomized smoothing, and please kindly let us know if you have any further questions.
>
> References:
> [1] A Unified Framework for Randomized Smoothing based Certified Defense. https://openreview.net/pdf?id=ryl71a4YPB
> [2] Filling the Soap of Bubbles: Efficient Black-Box Adversarial Certification with Non-gaussian smoothing. https://openreview.net/pdf?id=Skg8gJBFvr

---

### Official Review · AnonReviewer3 · 2019-10-26
**Official Blind Review #3**

**Rating:** 3

**Review:**

This paper proposes a new variation on certified adversarial training method that builds on two prior works IBP and CROWN. They showed the method outperformed all previous linear relaxation and bound propagation based certified defenses.

Pros:
1. The empirical results are strong. The method achieved SOTA.

Cons:
1. Novelty seems small. It is a straightforward combination of prior works, by adding two bounds together.
2. Adds a new hyperparameter for tuning.
3. Lack of any theoretical insights/motivation for the proposed method. Why would we want to combine the two lower bounds? The reason given in the paper is not very convincing:

"IBP has better learning power at larger epsilon and can achieve much smaller verified error.
However, it can be hard to tune due to its very imprecise bound at the beginning of training; on the
other hand, linear relaxation based methods give tighter lower bounds which stabilize training, but it
over-regularizes the network and forbids us to achieve good accuracy."

My questions with regards to this:
(i) Why does loose bound result in unstable training? Tighter bound stabilize training?
(ii) If we're concerned with using a tighter bound could result in over-regularization, then why not just combine the natural loss with the tight bound, as natural loss can be seen as the loosest bound. Is IBP crucial? and why?


**Experience Assessment:**

I do not know much about this area.

**Review Assessment: Checking Correctness Of Derivations And Theory:**

I assessed the sensibility of the derivations and theory.

**Review Assessment: Checking Correctness Of Experiments:**

I assessed the sensibility of the experiments.

**Review Assessment: Thoroughness In Paper Reading:**

I read the paper at least twice and used my best judgement in assessing the paper.

---

> ### Author Response · Authors · 2019-11-09
> **More explanations on the bounds and why tight bounds can stabilize training**
>
> Dear AnonReviewer3,
>
> Thank you for providing your helpful feedback. Sorry for the potential confusion in our paper and we would like to clarify them in our response, and we have added relevant discussions to our revision.
>
> (Q1) Why does loose bounds result in unstable training? Tighter bounds stabilize training?
>
> Loose bounds (lower bounds on margins $\underline{m}$, as defined on page 4 “verification specifications”) give a loose upper bound of the minimax loss (Eq. 4); in other words, the “robust loss” term in (9) will become very large. A large robust loss can be a challenge for the optimizer to minimize, and training can easily diverge or stuck at random guess level.
>
> More specifically, we will explain why a tighter bound like CROWN-IBP can help to stabilize IBP training.
>
> When taking a randomly initialized network or a naturally trained network, IBP bounds are very loose. But in Table 1, we show that a network trained using IBP can eventually obtain quite tight IBP bounds and high verified accuracy (i.e., the network can adapt to IBP bounds and learn a specific set of weights to make IBP tight and also correctly classify examples). However, since the training has to start from weights that produce loose bounds for IBP, the beginning phase of IBP training can be challenging and is vitally important.
>
> We observe that IBP training can have a large performance variance across models and initializations. Also, IBP is more sensitive to hyper-parameters like kappa or schedule length; as you can see in Figure 3, many IBP models failed to converge (large worst/median verified error) with some kappa or schedule length settings. The reason for instability is that during the beginning phase of training, the loose bounds produced by IBP make the robust loss (Eq. 9) explode, and it is challenging for the optimizer to reduce this loss and find a set of good weights that produce tight IBP verified bounds in the end.
>
> Conversely, if our bounds are much tighter at the beginning, the robust loss (Eq. 9) always remains in a reasonable range during training, and the network can gradually learn to find a good set of weights that make IBP bounds increasingly tighter. Initially, tighter bounds can be provided by a convex relaxation-based method, and the convex relaxation bounds are gradually replaced by IBP bound (using beta_start=1, beta_end=0), eventually leading to a model with learned tight IBP bounds in the end.
>
> To give you some intuitions on how much tighter CROWN-IBP is than IBP, in appendix B, we added a figure comparing the tightness between IBP bound and CROWN-IBP bound. We take the difference between the two bounds (CROWN-IBP bound minus IBP bound) and plot this difference during the training procedure. At the beginning of training, the bound difference can be very large, and the network gradually learns how to make the IBP bounds tighter during the training process. The use of tighter bounds at the beginning prevents divergence and can guide the network to learn better IBP bounds and achieve better-verified accuracy.
>
> (Q2) why not just combine the natural loss with the tight bound, as natural loss can be seen as the loosest bound? Is IBP crucial? and why?
>
> The natural loss does not provide a bound. The loosest bound is negative infinity, and it will explode the robust loss in Eq. (9). (see page 4, verification specifications, for the definition of this lower bound on margin, and Eq (4) for how to use it to get an upper bound for the minimax loss).
>
> IBP is crucial because it can provide us with a bound for computing the robust loss. Natural training cannot provide such bounds. Additionally, a network trained using natural loss is not robust and difficult to verify with current techniques. Models trained using IBP bounds as $\underline{m}$ in Eq. (9) can be quickly verified using IBP.
>
> --to be continued

---

> > ### Author Response · Authors · 2019-11-09
> > **Continue**
> >
> >
> > (Q3) Why would we want to combine the two lower bounds? (“Lack of any theoretical insights/motivation for the proposed method”)
> >
> > The paper was motivated by the success and weakness of IBP. Gowal et al. showed that with careful tuning, IBP can significantly outperform convex relaxation-based defense (e.g., on MNIST epsilon=0.3 it reduces verified error from 40% to <10%). We believe part of the reason is that convex relaxation-based methods overregularize the network in large epsilon regime, as we observe in Figure 1. However, in practice, IBP training can be unstable, as shown in Figure 3, where IBP cannot converge very well in all settings. The reason for unstable training is due to the loose bound at the beginning of training, as we have explained in (Q1) above. Additionally, IBP is computationally very cheap (so bounds are loose) but convex relaxation-based methods are in the order of 100X more expensive (see Table G) to obtain tighter bounds.
> >
> > The combination of the two bounds helps us get the best of both worlds (which is also mentioned by AnonReviewer1): tighter bounds at the beginning of training so more stable training (see Q1 above); no overregularization since we can gradually decay the convex relaxation bounds when epsilon increases; much better computational efficiency compared to regular convex relaxation-based methods since we reuse IBP bounds for intermediate layers.
> >
> > Both convex relaxation-based methods (see Salman et al., 2019 for an overview) and IBP are theoretically sound bounds, the combination of them are still sound and within the minimax optimization framework (Eq. 2). However, we agree that there is no theoretical answers to the questions why IBP works better than convex relaxations for large perturbation radii, and why convex relaxation-based approach over-regularize. We discussed a possible hypothesis in Appendix A, however, we believe this is still an open challenge and beyond the scope of our paper.
> >
> >
> > (Q4) Novelty seems small
> >
> > Our paper is the first to propose such a unique combination of convex relaxation and IBP based verifiable training procedure, and our empirical results achieve state-of-the-art, outperforming all baseline methods in all settings significantly on MNIST and CIFAR-10.
> >
> > Most importantly, our paper is not a naive combination of two methods. There are lots of rationales behind the scenes:
> >
> > 1. As discussed in (Q1) and (Q3), we carefully considered the strengths and weaknesses of both IBP and convex relaxation-based method, provided empirical studies (Table 1 and Figure 1) and designed our method to exploit the strengths of both methods. It is not a naive combination of two random methods; this combination has a clear justification: improve stability at the beginning phase of training and avoid overfitting at the late phase of training.
> >
> > 2. The computational cost of the convex relaxation-based method is typically very high. A naive combination will not overcome the drawback of high computational complexity. CROWN-IBP cleverly avoids this problem by re-using IBP bounds as intermediate layer results for convex relaxation, reducing the complexity of convex relaxation-based method from $O(L^2 n^3)$ to $O(L n^2 n_L)$ where usually $n_L$ is much less than $n$ (see the “Computational Cost” paragraph on page 7). The reduction of computation is in the order of 100 (see Table G, training time comparison).
> >
> > 3. Additionally, CROWN-IBP allows efficient implementation for convolutional networks on accelerators (GPUs/TPUs), because the backward bound propagation pass always starts from the last specification vector, which is guaranteed to be a small dense matrix. (See page 7, paragraph “Computational Cost” for detailed discussions).
> >
> > --to be continued

---

> > > ### Author Response · Authors · 2019-11-09
> > > **Continue**
> > >
> > >
> > > (Q5) Adds a new hyperparameter for tuning
> > >
> > > Compared to IBP, CROWN-IBP only adds one additional hyperparameter, $\beta$. $\beta$ has a clear meaning: balancing between the convex relaxation-based bounds and the IBP bounds.
> > >
> > > In all our experiments, we did not tune or search this hyperparameter. We fix $\beta_{start}=1$, $\beta_{end}=0$ for all experiments (except for CIFAR 2/255), and this allows us to achieve SOTA. In the CIFAR 2/255 setting, it is known that convex relaxation-based methods outperform IBP, thus it is intuitive to set $\beta=1$.
> > >
> > > We added a new paragraph “Hyperparameter κ and β” to the end of appendix C, giving more insights on how to select these hyperparameters. $\beta$ is a hyperparameter that rarely needs to be tuned, and when you change it you know what to expect. It is not a blackbox hyperparameter that requires a grid search or luck to choose the best value. We believe the addition of this hyperparameter is not a significant con of CROWN-IBP.
> > >
> > > (Conclusions)
> > >
> > > We hope we have addressed the concerns of the reviewer, especially why we combine the two bounds and why loose bounds result in unstable training. Despite being simple at the first glance, there are no existing works of this kind and our methods take a lot of design considerations (bound tightness, avoiding over-regularization, computational efficiency) behind the scenes and also achieves state-of-the-art performance. We hope the reviewer can re-evaluate our paper based on the response, and we look forward to discussing it with the reviewer if further concerns are raised.

---

> ### Author Response · Authors · 2019-11-13
> **Thank you for your review! We hope you can read our response soon.**
>
> Dear AnonReviewer3,
>
> Thank you again for your constructive review. Since the discussion period is closing soon, we will really appreciate it if you can read our response and provide us some feedback. We will be glad to discuss with you on any further concerns.
>
> In our response, we have discussed in detail why a tight bound is helpful for training, and the rationales for combining the two bounds behind the scenes. Our paper is not a naive combination of the two bounds, and we took careful design considerations to get the best of both worlds and achieve SOTA. We hope the reviewer can understand our paper better after reading our response.
>
> Thank you,
> Paper 1473 Authors

---

### Official Review · AnonReviewer1 · 2019-11-04
**Official Blind Review #1**

**Rating:** 8

**Review:**

This work proposes CROWN-IBP - novel and efficient certified defense method against adversarial attacks, by combining linear relaxation methods which tend to have tighter bounds with the more efficient interval-based methods. With an attempt to augment the IBP method with its lower computation complexity with the tight CROWN bounds, to get the best of both worlds. One of the primary contributions here is that reduction of computation complexity by an order of \Ln while maintaining similar or better bounds on error. The authors show compelling results with varied sized networks on both MNIST and CIFAR dataset, providing significant improvements over past baselines.

The paper itself is very well written, lucidly articulating the key contributions of the paper and highlighting the key results. The method and rationale behind it quite easy to follow.


Pros:
> Show significant benefits over previous baseline with 7.02% verified test error on MNIST at  \epsilon = 0.3, and 66.94% on CIFAR-10 with \epsilon = 8/255
> The proposed method is computationally viable, with up to 20X faster than linear relaxation methods with similar. better test errors and within 5-7X slower than the conventional IBP methods with worse errors

Cons:
> Extensive experiments with more advanced networks/datasets would have been more convincing, esp. given the computation efficiency that enables such experiments
> More elaborate insights into the choice of the training config/hyper-params esp. with the choice of \K_start, \K_end across the different datasets


Other comments:
> For the computational efficiency studies, it would be helpful to provide a breakdown of the costs between the different layers and operations, to better asses/confirm that benefits of CROWN-IBP method
> Impact of other complementary techniques such a lower precision/quantization? One fo the references compared against is the Gowal et al. 2018 for the as a baseline, however, it seems to be those results were obtained on a different HW platform (TPUs - motioned in Appendix-B), with potentially different computational accuracies (BFLOAT16 ?). So, this bears to question of the impact of precision on these methods and also the computation complexity.
> Since I'm not very well versed with the current baseline and state-of-art for variable robust training of DNN, it would be good to get an additional confirmation on the validity of the used baselines.

**Experience Assessment:**

I do not know much about this area.

**Review Assessment: Checking Correctness Of Derivations And Theory:**

I assessed the sensibility of the derivations and theory.

**Review Assessment: Checking Correctness Of Experiments:**

I assessed the sensibility of the experiments.

**Review Assessment: Thoroughness In Paper Reading:**

I read the paper at least twice and used my best judgement in assessing the paper.

---

> ### Author Response · Authors · 2019-11-09
> **Thank you for the encouraging comments. We have added additional results, and further speedup implementation.**
>
> We thank the reviewer for the encouraging comments and constructive feedback. We really appreciate the reviewer’s precise characterization of the contributions in our work. We provide answers to the raised questions/cons below.
>
> (Q1). Extensive experiments on advanced networks/datasets
>
> In our paper we use the same networks as the previous work (Gowal et al. 2018), to stay comparable with their results, which we call DM-Small, DM-Medium, and DM-Large. During the preparation of this submission, we tried much wider networks by increasing the width of the DM-Large model twice and four times, but they did not yield significant performance improvement. Thus we decided to keep models the same as in previous work for a straightforward comparison. We plan to implement more advanced networks (e.g., ResNet, DenseNet, etc) as the next step and scale to larger datasets is our future work.
>
> Besides the three models presented in the submitted version of the paper, in this revision, we additionally provide more comprehensive experiments on a large range of MNIST and CIFAR-10 models (18 MNIST models + 17 CIFAR models). The purpose of this experiment is to compare model performance statistics (min, median, and max) on a wide range of models, rather than a few hand-selected models. The results are presented in Appendix F, Table D. On all model structures and parameter settings, CROWN-IBP can outperform IBP in terms of best, median and worst verified errors. Especially, in many situations, the worst-case verified error improves significantly using CROWN-IBP because IBP training is not stable on some of the models.
>
> (Q2). More elaborate insights into the choice of the training config/hyper-params:
>
> This is a very good suggestion. kappa controls the trade-off between verified accuracy and standard (clean) accuracy and we typically recommend kappa_start=1 and kappa_end=0. beta determines if we want to use a convex relaxation-based the bound or IBP based bound; the general recommendation is to set beta_start=1 and beta_end=0. We added three paragraphs at the end of Appendix B to discuss the selection of hyperparameters in detail.
>
> (Q3). Breakdown of the cost between different layers and operations:
>
> The per-layer cost for propagating the CROWN-IBP bound backward is actually quite simple: in a high level, for all operations in the neural network, it is $n_L - 1$ times ($n_L$ is the number of classes, for MNIST/CIFAR it is 10) more expensive than forward propagation as there are $n_L - 1$ specifications per example. Thus CROWN-IBP is well suited to problems where the number of classes is small (more classes can be done efficiently by subsampling of specifications, which is left to future work). CROWN-IBP is significantly more efficient than CROWN (Zhang et al., 2018) and convex adversarial polytope (Wong et al., 2018); it is $L n$ times faster than these approaches, where $L$ is the number of layers and $n$ is hidden layer size. Generally, (ordinary) CROWN and convex adversarial polytopes are too slow for training.
>
> Practically, CROWN-IBP training time can be much less than $n_L - 1$ times slower than IBP, as CROWN-IBP is typically only used during the epsilon schedule rather than the entire training process, and CROWN-IBP generally executes more efficiently than IBP on parallel hardware because it packs denser computation that utilizes hardware accelerators better.
>
> Empirically, we have further optimized the implementation of CROWN-IBP (with roughly 2X reduction in training time in Table G), and we have prepared a multi-GPU version that can train the largest CIFAR-10 model in about 1 day using 4 GPUs. We have provided updated training time measurement in Appendix J and Table G. On the largest CIFAR-10 model, training using CROWN-IBP is actually only about twice slower than IBP.
>
>
> (Q4). Complementary techniques such as lower precision/quantization:
>
> To see if bfloat16 has any impact on training results, we additionally implement CROWN-IBP on multi-GPUs with float32. We train the CIFAR-10 model using the same hyperparameters as on TPUs and we found that the differences between TPU and GPU training results are small. The results are provided in Table H. We see no big difference between bfloat16 and float32 training.
>
> (Q5). Confirmation on state-of-the-art verifiable training baseline:
>
> For the verification of L infinity norm perturbations, the current best baselines in most settings are IBP (Gowal et al., 2018), except on CIFAR 2/255 where (Wong et al. 2018) is the best. CROWN-IBP can achieve better-verified accuracy than previous state-of-the-art works in all settings.
>
> We thank the reviewer again and will be glad to discuss with the reviewer on any parts that are still unclear, or any additional concerns raised.

---

### Public Comment · ~Matthew_B_Mirman1 · 2019-10-03
**Characterization of Related Work**

I would like to address a couple of points:

1. In the related section on page 3, you say:

> Mirman et al. (2018) proposed to use “Hybrid Zonotope Domain” which is effectively IBP to scale up linear relaxation based training. Gowal et al. (2018) first demonstrated that IBP could outperform many state-of-the-art results by a large margin after careful tuning.

This is a poor characterization.   Mirman et al (ICML July, 2018) proposed to use a variety of domains, including the Box/Interval domain (IBP) and the Hybrid Zonotope Domain (unrelated to IBP in any form).  This paper showed that IBP could scale to much larger networks then other works existing at the time could (Raghunathan, et al, 2018) and was concurrent with Wong & Kolter (ICML 2018).  Gowal et al. (first released October, 2018, published in ICCV 2019) used interval (IBP) but with more precise approximations than interval for the last linear layer of the network and used linear parameter annealing in the training scheme to achieve even better results.


2.  In "Issues with linear relaxation based training." you say:

> DiffAI (Mirman et al., 2018), is their high computational and memory cost, and poor scalability.

In fact, DiffAI introduced the first IBP based training system, and thus it makes no sense to claim that it is much slower and more memory consumptive than other techniques which are also built using IBP.

For more metrics of the efficiency and performance using the DiffAI framework, see https://arxiv.org/pdf/1903.12519.pdf and the reproducible numbers in the DiffAI repo: https://github.com/eth-sri/diffai

---

> ### Author Response · Authors · 2019-10-03
> **Thank  for the comments. We will address your concerns on related works in our revision.**
>
>
> Dear Matthew,
>
> Thank you for your comments on related works and they are very helpful for improving our paper!
>
> 1. Apologies for the confusion and we agree it is inaccurate. In fact, we actually meant “Box Domain” rather than “Hybrid Zonotope Domain”; I accidentally typed the wrong word. According to your comments, we will revise the sentence to the following:
>
> Mirman et al. (ICML 2018) proposed a variety of abstract domains for bound propagation, including the “Box/Interval Domain” (IBP) and showed that it could scale to much larger networks than other works (Raghunathan, et al, 2018) could at the time. Gowal et al. (ICCV 2019) demonstrated that IBP could outperform many state-of-the-art results by a large margin with more precise approximations for the last linear layer and better training schemes.
>
> Does this look good to you? Let us know if you have any other concerns.
>
> 2. Thank you for pointing this out. We will remove DiffAI from that sentence. Our main concern is on the full convex relaxation-based training method like Wong & Kolter (ICML 2018), and it is inaccurate the include DiffAI in that sentence. We will revise any other possibly inaccurate citations as well.
>
> 3. Thank you for pointing us to your interesting work (arxiv 1903.12519) demonstrating the latest performance of the versatile DiffAI framework. We will cite this paper and include the best numbers reported in your paper into our main result table (Table 2).
>
> We appreciate your helpful comments and please kindly let us know if you have any additional concerns.
>
> Sincerely,
> Paper 1473 authors

---

> > ### Public Comment · ~Matthew_B_Mirman1 · 2019-10-04
> > **Improved**
> >
> > Yes, this looks better.  However, I'd note that the phrase "bound propagation" has caused confusion in the past and led to the misunderstanding that these domains only track bounds per neuron or per layer. At their core, more domains such as HZD and DeepPoly keep track of relationships between neurons and even between layers (the necessity of which becomes evident for residual networks).

---

> > > ### Author Response · Authors · 2019-10-05
> > > **Thanks for the feedback; we made it more clear**
> > >
> > > Hi Matthew,
> > >
> > > Thanks for your feedback! And yes we agree that the phrase "bound propagation" may cause confusion and does not accurately describe the work (Mirman et al. 2018). To make things clear, we will change the beginning of this sentence to:
> > >
> > > Mirman et al. (ICML 2018) proposed a variety of abstract domains to give sound over-approximations for neural networks, including the “Box/Interval Domain” (referred to as IBP in Gowal et al.) and showed that…
> > >
> > > Hope it is even better now. Thanks again for your helpful comments!
> > >
> > > Sincerely,
> > > Paper 1473 Authors

---

### Author Response · Authors · 2019-10-21
**Erratum: CROWN-IBP verified error in Table 2 for MNIST epsilon_test=0.4 should be 12.06%, not 12.84%**

In Table 2 of submitted draft, the best reported verified error for MNIST epsilon_test=0.4 is 12.84%. This number does not match other places in paper (in introduction and Appendix B, page 14, we mentioned 12.06%). During final formatting, the 12.84% number in Table 2 was mistakenly copied from results using a wrong training schedule. The correct verified error, clean error and PGD error for MNIST epsilon_test=0.4 are 12.06%, 2.17% and 9.47% respectively; both verified and clean error are noticeably better. We will make these corrections to Table 2 in our revision.

Thanks,
Paper1473 Authors

---

### Author Response · Authors · 2019-11-15
**Summary of changes and new results**

Dear Reviewers and Area Chair,

We summarize the changes we made to address reviewer concerns, and the new results added to the revision of our paper as follows.

1. The use of TPUs to achieve SOTA results may be a concern, so we have implemented a multi-GPU version of CROWN-IBP. To train the SOTA CIFAR model, it takes 1 day on 4 GPUs and the obtained verified accuracy is similar to the results obtained on TPUs. Results are provided in Appendix G, Table H. We will open source our multi-GPU training code.

2. We have improved the implementation of CROWN-IBP. Running time measurement is provided in Appendix J, Table G. For the largest network, CROWN-IBP is just about twice slower than IBP (both are trained on 4 GPUs). We also discussed these results in Appendix J.

3. We have included performance evaluation on a large amount of smaller models trained on a single GPU (in Table C, Appendix F) by investigating performance statistics (min/median/max). This prevents hand-tuning on a small set of models, and we can consistently outperform IBP over a large range of models.

4. We have added more discussions on why a tighter bound can stabilize IBP training in Appendix B and bound tightness comparison between IBP and CROWN-IBP (Figure B).

5. We add more discussions on how to choose hyperparameters in Appendix C. Compared to IBP, CROWN-IBP only adds one additional hyperparameter, $\beta$. We did not tune or grid-search this hyperparameter in any experiments; $\beta$ has a clear meaning and our selection has clear justifications.

We thank the reviewers again for their helpful comments and hope they can re-evaluate our paper based on our response to each reviewer, and the highlighted updates in our paper above.

Thanks,
Paper 1473 Authors

---

### Decision · Program_Chairs · 2019-12-19

**Decision:**

Accept (Poster)

**Comment:**

This paper presents a method that hybridizes the strategies of linear programming and interval bound propagation to improve adversarial robustness.  While some reviewers have concerns about the novelty of the underlying ideas presented, the method is an improvement to the SOTA in certifiable robustness, and has become a benchmark method within this class of defenses.